**Turbulence and hypoxia contribute to dense biological scattering layers in**
**Patagonian Fjord System.**
**Authors:** Iván Pérez-Santos[1-2*], Leonardo Castro[2-3-4], Lauren Ross[5], Edwin Niklitschek[1],
Nicolás Mayorga[1], Luis Cubillos[2-3], Mariano Gutierrez[6], Eduardo Escalona[2-3], Manuel
Castillo[2-7], Nicolás Alegría[8] and Giovanni Daneri[2-9].
[1]Universidad de Los Lagos, Centro i~mar, Camino a Chinquihue km 6, Puerto Montt, Chile.
[2]COPAS Sur-Austral, Universidad de Concepción, Campus Concepción, Víctor Lamas 1290,
Casilla 160-C, código postal: 4070043, Concepción, Chile.
[3]Departamento de Oceanografía, Universidad de Concepción, Campus Concepción, Víctor
Lamas 1290, Casilla 160-C, código postal: 4070043, Concepción, Chile.
[4]Centro de Investigaciones de Altas Latitudes (IDEAL), Universidad Austral de Chile,
Valdivia, Chile.
[5]Department of Civil and Environmental Engineering, University of Maine, 5711 Boardman
Hall, Orono, ME 04469-5711.
[6]Universidad Nacional Federico Villareal, Facultad de Oceanografía, Pesquerías y Ciencias
Alimentarias, Calle Francia 726, Miraflores, Lima, Perú.
[7]Centro de Observación Marino para Estudios de Riesgo en Ambientes Costeros, Facultad de
Ciencias del Mar y de Recursos Naturales, Universidad de Valparaíso, Chile.
[8]Instituto de Investigaciones Pesqueras, Talcahuano, Chile.
[9]Centro de Investigaciones en Ecosistemas de la Patagonia (CIEP), Coyhaique, Chile.
***Corresponding author**: Iván Pérez-Santos, email: ivan.perez@ulagos.cl

## Abstract

The aggregation of plankton species along fjords can be linked to physical properties and processes such as stratification, turbulence and oxygen concentration. The goal of this study is to determine how water column properties and turbulent mixing affect the horizontal and vertical distributions of macrozooplankton along the only north Patagonian Fjord known to date where hypoxic conditions occur in the water column. Acoustic Doppler Current Profiler moorings, scientific echo-sounder transects, and *in-situ* plankton abundance measurements were used to study macrozooplankton assemblages and migration patterns along Puyuhuapi Fjord and Jacaf Channel in Chilean Patagonia. The dissipation of turbulent kinetic energy was quantified through vertical microstructure profiles collected throughout time in areas with high macrozooplankton concentrations. The acoustic records and *in-situ* macrozooplankton data revealed diel vertical migrations (DVM) of siphonophores, chaetognaths and euphausiids. In particular, a dense biological backscattering layer was observed along Puyuhuapi Fjord between the surface and the top of the hypoxic boundary layer (~100 m), which limited the vertical distribution of most macrozooplankton and their DVM, generating a significant reduction of habitat. Aggregations of macrozooplankton and fishes were most abundant around a submarine sill in Jacaf Channel. In this location macrozooplankton were distributed throughout the water column (0 to ~200 m), with no evidence of a hypoxic boundary due to the intense mixing near the sill. In particular, turbulence measurements taken near the sill indicated high dissipation rates of turbulent kinetic energy ($\varepsilon$ ~$10^{-5}$ W kg$^{-1}$) and vertical diapycnal eddy diffusivity ($K_\rho$ ~$10^{-3}$ m$^2$ s$^{-1}$). The elevated vertical mixing ensures that the water column is well oxygenated (3-6 mL L$^{-1}$, 60-80 % saturation), creating a suitable environment for macrozooplankton and fish aggregations. Turbulence induced by tidal flow over the sill apparently enhances the interchange of nutrients, oxygen concentrations, and create a fruitful environment for many marine species, where prey-predator relationship might be favored.

**Keywords:** turbulence, hypoxia, Acoustic data, macrozooplankton, scientific echo-sounder, Patagonian fjords, sill exchange.

## 1 Introduction

Spatial and temporal variability of plankton assemblages have been linked to oceanographic features and processes such as water column stratification, tidal mixing and turbulence, frontal structures, advection, and secondary circulation in estuaries and fjords (Govoni et al., 1989; Rodriquez et al., 1999; Lee et al., 2005; Lough and Manning, 2001; Munk et al., 2002; Meerhoff et al., 2013; Meerhoff et al., 2015). In fjords, residual flows resemble typical estuarine gravitational circulation with landward flow at depth and seaward flow at the surface. It has been found that residual flows in fjords can retain planktonic larvae inside the system (Dyer, 1997; North and Houde, 2001, 2004; Meerhoff et al., 2015). Another recent study has shown that advection can influence the import and export of zooplankton in a fjord depending on the depth at which the zooplankton are located, which can potentially affect the community composition, biomass, productivity and distribution of zooplankton in the fjord (Basedow et al., 2004). Moreover, horizontal mixing of along-channel density gradients has been shown to induce lateral circulation (Farmer and Feeland, 1983), which in turn affects larval distributions in fjord systems (Meerhoff et al., 2015).

Other recent studies have investigated how tidally asymmetries in mixing, and thus tidal variations in stratification, affects ichthyoplankton and zooplankton assemblages (Pérez et al., 1977; Nixon et al. 1979; Oviatt, 1981, Lee et al., 2005). Lee et al., (2005) found that variations in stratification throughout a tidal cycle affected both overall abundance and species composition of zooplankton in the Irish Sea. However, they did not have the tools to relate the hydrodynamic and hydrographic variability of this region to vertical and horizontal distributions of fish larvae and zooplankton. Another study by Oviatt (1981) found that zooplankton concentrations were lower in laboratory tanks than in nature (Narragansett Bay, USA). Since this was not due to the physical action of mixing (induced by paddles in the tank), they hypothesized that tank confinement and turbulence had broken down vertical segregation between adults and juveniles, resulting in increased cannibalism. While vertical segregation of zooplankton groups, probably related to different trophic guilds, has been confirmed by several studies (e.g. Haury et al., 1990), this segregation can be reduced by turbulent processes enhancing contact between prey and predators (Visser and Stips 2002; Visser et al., 2009). For instance, available theoretical models predict optimal prey consumption at dissipation rates of turbulent kinetic energy ($\varepsilon$) between $10^{-6}$ and $10^{-4}$ W kg$^{-1}$ (Lewis and Pedley, 2001). In fjords, topographic conditions are extremely irregular (Inall and

Gillibrand, 2010), inducing high turbulence and enhanced vertical mixing, particularly at sills
(Klymak and Gregg 2004; Whitney et al., 2014). However, enhanced productivity,
oxygenation, and/or local retention may occur at these same highly turbulent areas. For
example, turbulence is known to mix freshwater inflow with deep, dense ocean water,
allowing for oxygenation of the deeper layers (MacCready et al., 2002; Peters and Bokhorst,
2001) and turbulent these eddies can impact phytoplankton bloom growth (Cloern, 1991;
Koseff et al., 1993). Therefore, additional field studies are needed to confirm the relationship
between mixing-inducing physical forcing, such as wind or advection, and vertical abundance
patterns and species composition in fjords and other estuarine systems. One of the principal
questions that will be address in this study is: what is the contribution of turbulence to the
mixing of fjord water column properties (e.g., Temperature, salinity and dissolved oxygen)
and to the aggregation of macrozooplankton at certain depths (scattering layers) along north
Patagonian fjords and channels, emphasizing the role of sills in some locations (e.g., Jacaf
Channel, Fig. 1)?

Dissolved oxygen (DO) is the most important dissolved gas in the ocean as it sustains

marine life and ensures ecosystems health. Most of the world's oceans are oxygenated,
however there are some regions of low DO, referred to as hypoxic zones and if their DO
concentrations are equal or close to 0 mL L$^{-1}$ they are known as "Dead zones" (Díaz et al.,
2001; Ekau et al., 2010; Hauss et al., 2016). Throughout the world's oceans there exist areas
where hypoxic conditions are permanent and where the DO is significantly lower than well-
oxygenated areas (e.g., <20 µM or 0.31 mL L$^{-1}$). These areas are known as Oxygen Minimum
Zones (OMZs) and due to the upwelling associated with them; they experience elevated
primary production at the surface (Mass et al., 2014; Hauss et al., 2016; Seibel et al., 2016).
The major ocean OMZs are located in the Eastern South and North Pacific, the Arabian Sea,
Bay of Bengal (Indian Ocean), West Bering Sea, the Gulf of Alaska and the Eastern North
Atlantic, covering around 8% of the total ocean (~30 million km$^2$) (Paulmier and Ruiz-Pino,
2009; Fuenzalida et al., 2009; Hauss et al., 2016). The Eastern South Pacific OMZ (ESP-
OMZ), present along the Chilean coast, represents an area of 9.8 million km$^2$ (2.6 % of the
total ocean) (Fuenzalida et al., 2009). Even the ESP-OMZ decreased and disappeared south of
~37° S, however water with low DO (2-3 mL L$^{-1}$), associated with the Equatorial Subsurface
Water (ESSW), is still present up to 44° S (Silva et al., 2009). The ESSW water mass
infiltrates Patagonian fjords and channels and moves northward and southward (41.5°-44° S)
depending on the marine topography (Sievers and Silva, 2008).

Hypoxic conditions ($<$ 2 mL L$^{-1}$) have been detected in four regions of Patagonian

(Puyuhuapi Fjord, Jacaf Channel, Aysén Fjord and the Almirante Montt Gulf), and in each
region the oxygen depleted zones are mainly located at the fjords head and down to 100 m
depth (Silva and Vargas, 2014; Schneider et al., 2014). Some of the main contributors to
hypoxia in Patagonian fjords and channels have been found to be (1) water column
stratification causing separation between poorly oxygenated bottom water and oxygenated
surface waters, (2) DO consumption by degradation of organic matter (autochthonous and
allochthonous), (3) low ventilation due to the presence of deep bathymetric micro basins, (4)
advection of the ESSW and (5) anthropogenic activities such as industrial and sewage
discharge, riverine inputs of nutrients, agriculture activities, aquaculture, etc. (Sievers and
Silva, 2008; Silva and Vargas, 2014; Schneider et al., 2014).

Hypoxia is known to have a significant impact on plankton distribution and

development, hence on the health of the ecosystem as a whole (Ekau et al., 2010; Mass et al.,
2014; Hauss et al.,2016; Seibel et al., 2016). Some species can tolerate hypoxic water, e.g.,
smaller species, euphausiids and jellyfish can live in under 30% oxygen saturation and
dissolved oxygen of 1.6 mL L$^{-1}$. Other taxa, such as some copepods and fishes, may be more
sensitive to hypoxia and have preference for oxygen saturations of 50-100% and DO
concentrations of 2.6-5.2 mL L$^{-1}$ (Ekau et al., 2010; Mass et al., 2014; Hauss et al., 2016;
Seibel et al., 2016). The sensitivity of species to tolerate different oxygen concentrations,
however, may vary among organisms from different environments, e.g., coastal upwelling
zone, fjords systems and OMZ. Although hypoxic conditions have been detected in four
regions of Patagonia (Silva and Vargas, 2014; Schneider et al., 2014) no relationship has been
determined with the zooplankton species that inhabit this ecosystem. Therefore, the second
question that motivates this study is: How do hypoxic conditions affect the distribution and
aggregation of macrozooplankton species? This question will be addressed by investigating
Puyuhuapi Fjord and Jacaf Channel, two of the four hypoxic systems in Patagonia.

In Patagonian fjords, a comprehensive description of zooplankton distribution patterns

has been provided by Palma (2008), considering a total of 220 *in-situ* plankton samples, from
a number of depth strata between the surface and ~200 m. Main zooplankton groups included
siphonophores, chaetognaths, cladocerans, copepods and euphausiids. Although a positive

north to south gradient in the abundance of major zooplankton species was found, potential relationships between the vertical distributions and environmental variables were not deeply assessed. A later study by Landaeta et al., (2013) investigated the vertical distribution of microzooplankton and fish larvae in Steffen fjord (47.4° S) at four depth strata (200-50 m, 50-25 m, 25-10 m and 10-0 m depth) during November 2008. Copepod nauplii and copepodites of *Acartia tonsa* together with *Maurolicus parvipinnis* fish larvae were observed around the pycnocline region, suggesting that the vertical structure of the water column might play a role in larval fish distribution. More recently, studies on zoo- and ichthyoplankton vertical distributions in Reloncaví Fjord revealed that DVM timing might be modified by the tidal regime which is particularly strong in this area (Castro et al., 2014). However, none of these studies provided explicit assessments of the relationships between the vertical distribution of zooplankton and turbulent mixing or water column properties.

Most studies carried out in Chilean coastal waters, including those mentioned above, have relied on plankton nets and other collecting devices (pumps) deployed in single locations (fixed stations). An alternate approach is to use acoustic techniques, which can provide high resolution data on zooplankton DVM patterns (Valle-Levinson et al., 2014; Días-Astudillo et al., 2017) and segregation patterns throughout the water column (Sato 2013; Sato et al., 2016). For instance, DVM patterns of dense krill aggregations have been detected using Acoustic Doppler Current profilers (ADCP) moored around the Antarctic Peninsula, the Kattegat Channel and off Funka Bay, Japan (Buchholz et al., 1995; Lee et al., 2004; Zhou and Dorland 2004; Brierley et al., 2006). In Chilean fjords, ADCPs have been used to identify changes in vertical distribution and DVM patterns of zooplankton (e.g., from normal diel to twilight vertical migrations) over several months in Reloncaví Fjord (Valle-Levinson et al., 2014). These studies, although novel at describing temporal variations in zooplankton patterns, focused mainly on the behavior of particular species, but again did not consider how the vertical distribution of zooplankton is modified by water column conditions (e.g., temperature, salinity, oxygen and turbulence).

Compared to ADCPs, scientific echo-sounders are characterized by narrower beam angles, lower frequencies and longer ranges. They have also been used to provide valuable qualitative and quantitative information on various aquatic species and communities, from zooplankton to large predators (Ballón et al., 2011). Overall, macrozooplankton can be acoustically identified and virtually separated from other organisms, such as fish, by

considering their acoustic properties (Kloser et al., 2002; Logerwell and Wilson, 2004; Mosteiro et al., 2004; Simmonds and MacLennan, 2005). Although the use of several frequencies does not necessarily increase precision (Horne and Jech, 1999), the use of at least two frequencies (38 and 120 kHz) is currently a standard practice in zooplankton studies as identification methods developed by Ballón et al., (2011) and others can be utilized.

The present study aims to evaluate the effects of water column properties, such as dissolved oxygen and turbulent mixing, on the vertical distribution of dominant macrozooplankton groups along a Patagonian Fjord system. To achieve this goal ADCP and scientific echo-sounder data were combined with biological observations from *in-situ* stratified zooplankton samples and water column measurements from microstructure profilers and conductivity-temperature-depth-oxygen (CTDO) profilers. According to the information presented in this section, the principal hypotheses of this manuscript are: (1) the pervasive hypoxic layer existing in the Puyuhuapi Fjord limits DVM and overall distribution of macrozooplankton to the first 100 m depth of the water column, reducing the habitat of these species and (2) the higher turbulence originated by the tidal regime around sills favor the mixing of the water column, deepen the hypoxic layer, injects nutrients and, thus, increases primary production. Therefore, macrozooplankton exhibits higher densities and extends deeper in the water column around submarine sills.

## 2 Study Area

Patagonian fjords extend from 41° S to 56° S, and are typically deep and narrow as a result of their formation during glacial progression. Their hydrography is characterized by two vertical layers, consisting of a low salinity surface layer in the first ten meters of the water column (resulting from rainfall and glacial melt) that overlays a subsurface salty layer originated in the Pacific Ocean (Silva and Calvete, 2002; Pérez-Santos et al., 2014). Fjord systems play an important role in primary production and carbon cycling by providing a zone where energy and particulate material are exchanged between land and marine ecosystems (Gattuso et al., 1998). The principal nutrient (nitrate) is supplied to these fjords by oceanic transport, and particularly through the intrusion of Sub Antarctic Water (SAAW), a water mass that may also transport some species of zooplankton (González et al., 2011; 2013).

Puyuhuapi Fjord and Jacaf Channel are representative examples of the Patagonian
Fjord Systems. The main connection of Puyuhuapi Fjord with oceanic waters is via its
southern mouth. Although a second connection to oceanic water exists via Jacaf Channel,
interchange here is limited by the shallow Jacaf Channel sill, which is 50 m deep and 6 km
long. Its main freshwater input (the Cisnes River) meets the fjord half way between its head
and mouth (Fig.1). Jacaf Channel is well known for its great depth (> 400 m) around its
connection to the Moraleda Channel, which contrasts with its very shallow sill near its
connection with Puyuhuapi Fjord (Fig. 1). Seasonal hydrographic measurements along
Puyuhuapi Fjord have shown a stratified water column except in late winter, when the water
column became partially mixed due to a reduction in freshwater supply from rainfall and
glacial melting (Schneider et al., 2014). Hypoxic conditions have been detected in Puyuhuapi
Fjord below 100 m depth, where oxygen concentrations have been found to be as low as 1-2
mL $L^{-1}$ (Schneider et al., 2014; Pérez-Santos, 2017). This observed oxygen depletion could be
caused by limited ventilation due to shallow sills, or by the input of low-oxygen Equatorial
Subsurface Water into the fjord (Silva and Vargas, 2014; Schneider et al., 2014). Puyuhuapi
Fjord is the only northwestern Patagonian fjord known to experience such extreme hypoxic
conditions. At the same time, it is an area where intense aquaculture activities have been
recently developed, which reinforces the need of this study.

The study area offers an excellent opportunity for studying the impact of deep hypoxia
upon macrozooplankton distribution and behavior, considering the continued increase of
hypoxic regions around the world (Breitburg et al., 2018). Moreover, the presence of a sill in
Jacaf Channel, in the vicinity of its connection to the Puyuhuapi Fjord, opens the possibility
to investigate the influence of vertical mixing (Farmer and Freeland, 1983; Inall and
Gillibrand, 2010) upon water quality, especially upon dissolved oxygen concentration,
injection of nutrients from subsurface oxygen rich layers, enhancement of primary production
and, finally, upon the density of different zooplankton species (Pantoja et al., 2011).
Furthermore, the location of an oceanographic buoy in the northern part of Puyuhuapi Fjord
(Schneider et al., 2014) is a useful platform to carry out *in-situ* experiments combined with
oceanographic moorings.

**3 Data collection and methodology**
**3.1 Water column properties**
Hydrographic surveys were conducted during May and November 2013 and January and
August 2014 in Puyuhuapi Fjord and Jacaf Channel (Fig. 2, Table 1). These profiles were
obtained with a Seabird 25 CTDO, sampling at 8 Hz with a descent rate of ~1 m s$^{-1}$. The data
collected, whose nominal vertical resolution was ~12 cm, were averaged into 1 m bins,
following Seabird recommendations. The conservative temperature (˚C) and absolute salinity
(g kg$^{-1}$) were calculated according to the Thermodynamic Equation of Seawater 2010 (COI et
al., 2010). Additionally, nitrate samples were taken using a Niskin bottle at various depths and
analyzed spectrophotometrically following the methods of Strickland and Parsons (1968). To
validate CTDO oxygen measurements, *in-situ* oxygen samples were analyzed using the
Winkler method (Strickland and Parsons, 1968), carried out using a Metrohom burette
(Dosimat plus 865) and an automatic visual end-point detection (AULOX Measurement
System).

Microstructure measurements were collected using a Vertical Microstructure Profiler
(VMP-250, Rockland Scientific, Inc.). The VMP-250 is equipped with two airfoil shear
probes and two fast response FP07 thermistors, which allowed for data recording at 512 Hz
with a descending free fall speed of ~0.7 m s$^{-1}$. The micro-shear measurements permitted a
direct measurement of the dissipation rate of turbulent kinetic energy (ε) for isotropic
turbulence, according to Lueck et al., (2002), Eq. (1),
$$\varepsilon = 7.5\,\nu \overline{\left(\frac{\partial u'}{\partial z}\right)^2} \qquad (1)$$
where, ν is the kinematic viscosity, $u$ is the horizontal velocity, $z$ is the vertical coordinate
axis and therefore $\overline{\left(\frac{\partial u'}{\partial z}\right)^2}$ is the shear variance.

Using the values of ε, the diapycnal eddy diffusivity ($K_\rho$) was calculated. The most
used formulation was proposed by Osborn (1980),
$$K_\rho = \Gamma \frac{\varepsilon}{N^2}, \qquad (2)$$
where $\Gamma$ is the mixing efficiency, generally set to 0.2 (Thorpe 2005), and $N$ is the buoyancy
frequency.  Shih et al. (2005) noted that when the ratio $\varepsilon/\nu N^2$ is greater than 100, Eq. (2)
results in an overestimation.  Therefore, they proposed a new parameterization for this case
given by:
$$K_\rho = 2\nu \left(\frac{\varepsilon}{\nu N^2}\right)^{1/2}. \qquad (3)$$
More recently, Cuypers et al. (2011) used Eq. (3) when $\varepsilon/\nu N^2 > 100$, Eq. (2) when 7
$< \varepsilon/\nu N^2 < 100$, and considered null eddy diffusivity when $\varepsilon/\nu N^2 < 7$. This approach was
followed in this study. The correlation between the dissipation rate of turbulent kinetic energy
and the abundance of major zooplankton groups throughout the water column was
accomplished by using a quadratic polynomial curve fit between these data sets (explained in
detail in section 4.6). These analyses were only applied to measurements collected at the fixed
station in Puyuhuapi Fjord, because the VMP-250 was not available during the measurement
campaign in Jacaf Channel.

**3.2 Acoustic data**
Three types of acoustic data were collected: ADCP, single-frequency echo-sounder and dual-
frequency echo-sounder data. ADCP measurements were obtained with two 307.7 kHz
Teledyne RDI Workhorse ADCPs, moored upwards at depths of ~50 m (ADCP-1) and ~100
m (ADCP-2), both moored at the same location in north-central Puyuhuapi Fjord but during
different time periods (Table 1, Fig. 1). Data were collected hourly with a vertical bin size of
1 m, over periods of austral autumn (ADCP-1: May, 2013) and spring-summer (ADCP-2:
January 2014). During the final ADCP-2 mooring deployment, single-frequency data were
also collected along the Puyuhuapi Fjord using a SIMRAD EK60 scientific echo-sounder,
running a 38 kHz transducer (ES38B), during daytime and nighttime hours, from January 22-
25, 2014 (black line in Fig. 1). These ADCP and single-frequency echo-sounder
measurements were complemented by *in-situ* zooplankton sampling (see section 3.3 for
details) carried out on January 23-24, 2014, at a fixed station close to the ADCP mooring
location, over a period of 36 hours (Fig. 1).
A second scientific campaign was conducted on August 17[th] and 18[th], 2014, which
included a dual-frequency echo-sounder survey and a thrid ADCP mooring (ADCP-3) located
in Jacaf Channel. This time, the echo-sounder survey coverage was extended to eastern Jacaf
Channel (Fig. 1, red line) and a second 120 kHz transducer (ES120-7C) was added to the 38
kHz transducer used in the first survey. Several day/night transects were completed across
Puyuhuapi Fjord and Jacaf Channel, with special attention paid to Jacaf sill (only the most
representative echograms were showed in figures 5, 7 and 8). To determine the statistical
relationship ($R^2$) between acoustic data from the 38 kHz echo-sounder with hydrographic
properties of the fjords (temperature, salinity and dissolved oxygen), a quadratic polynomial

curve was also applied between these data sets. During this survey, two RDI Workhorse

ADCP was 614.4 kHz frequency (referenced hereafter as ADCP-3) and was moored at ~30 m

depth in the vicinity of the Jacaf sill. The near-surface placement of ADCP-3 allowed for

near-surface currents to be adequately quantified.

Vessel speed during all echo-sounder surveys was maintained between 8 and 10 knots. Echo-sounders were operated using a variable ping rate 0.3-2.0 ping s$^{-1}$, pulse duration of 1.024 milliseconds and output powers of 2 kW and 0.5 kW for the 38 and 120 kHz frequencies, respectively. Calibration was made using copper spheres and standard procedures (Foote et al., 1987).

### 3.2.1 Echo-sounder data analysis

Post-processing of echo-sounder data was performed in Echoview (Myriax inc, Tasmania, https://www.echoview.com/), where noisy data considered as those collected with weak pings, in blind areas, in the near field, with background noise or subjected to rainbow phenomenon were regarded as "bad data" and were eliminated. After this initial scrutiny and filtering step, all single-frequency echoes (38 kHz, Campaign 1) of intensity >-110 dB were considered and treated as a single "biological backscattering" class, which pooled all biological groups being present in the study area. Dual-frequency echoes, however, were classified into three different groups following Ballón et al., (2010). These authors built an algorithm, freely distributed as an Echoview template ("FishZpkPeru38&120.evi"), which uses both mean volume backscattering (MVBS) differences ($\Delta MVBS$) and summations ($\sum MVBS$) between 38 and 120 kHz to discriminate echoes into three different biological backscattering classes: fish and two macrozooplankton groups (macrozooplankton or "fluid-like" and gelatinous or "blue noise" organisms). The fluid-like group follows a sphere model (Holliday & Pieper, 1995) considered appropriate to represent cylindrical and spherical shapes, including euphausiids and large copepods, which are dominant macrozooplankton groups off Peru and Chile (Ayon et al., 2008). The algorithm is considered to be useful for 38 and 120 kHz data from targets whose radius is ≥0.5 mm and has a dB difference of 2-19 dB (Ballón et al., 2010 and 2011).

Given physical limitations imposed by near field and sound absorption effects related to the echo-sounder frequencies used (38 and 120 kHz), we defined and limited our analyses to an effective sampling range between 5 and 250 m. Absorption is greater for the 120 kHz

frequency, which exhibits the shortest range, but has a greater vertical resolution than 38 kHz echo-sounder. The 38 kHz frequency, on the other hand, exhibits a much longer range (>1000 m), but limited resolution regarding small zooplankton scatterers., It has been shown, however, to be efficient for studying macrozooplankton distributions of siphonophores, chaetognaths and euphausiids (Mair et al., 2005; Cade and Benoit-Bird, 2015; Ariza et al., 2016).

Volume backscattering strength ($S_v$, dB re 1 m$^{-1}$) values from the single-frequency and from each of the three dual-frequency virtual echograms were integrated and re-scaled into the customary index "nautical area scattering coefficient" (NASC, in units of m$^2$ n mi$^2$), using a grid of 20 m (depth) by 50 m (distance). Since NASC lies on the linear domain, it can be considered proportional to and suitable for indexing targets abundance (Ballón et al., 2011).

Quadratic polynomial models were fit to assess the statistical relationship ($R^2$) between biological scattering (single-frequency integrated data) and the hydrographic variables measured in each fjord (temperature, salinity and dissolved oxygen).

### 3.2.2 Acoustic data analysis from ADCPs

ADCP echo intensity was converted to mean volume backscattering strength ($S_v$, dB re 1 m$^{-1}$), as done for scientific echo-sounder data, following the conversion formula:

$$S_v = C + 10log[(Tx + 273.16)R^2] - L_{DBW} - P_{DBW} + 2\alpha R + K_c(E - E_r) \qquad (4)$$

where, $C$ is a sonar-configuration scaling factor (-148.2 dB for the Workhorse Sentinel), $T_x$ is the temperature at the transducer (°C), $L_{DBW}$ is log$_{10}$(transmit-pulse length, $L$=8.13 m), $P_{DBW}$ is log$_{10}$(output power, 15.5 W), $\alpha$ is the absorption coefficient (dB m$^{-1}$), $K_c$ is a beam-specific sensitivity coefficient (supplied by the manufacturer as 0.45), $E$ is the recorded AGC (automatic gain control), and $E_r$ is the minimum AGC recorded (40 dB for ADCP-1 and 41 dB for ADCP-2). The beam-average of the AGC for the 4 transducers was used to obtain optimal results following the procedure in Brierley et al. (2006). Finally, $R$ is the slant range to the sample bin (m), which uses the vertical depth as a correction (Lee et al., (2004)). Therefore, $R$ is expressed as,

$$R = \frac{b + \frac{L+d}{2} + ((n-1)d) + (d/4)}{\cos\zeta} \frac{\bar{c}}{c_I} \qquad (5)$$

where $b$ is the blanking distance (3.23 m), $L$ is the transmit pulse length (8.13 m), $d$ is the length of the depth cell (1 m), $n$ is the depth cell number of the particular scattering layer being measured, $\zeta$ is the beam angle (20°), $\bar{c}$ is the average sound speed from the transducer to

the depth cell (1453 m s$^{-1}$) and $c_I$ is the nominal sound speed used by the instrument (1454 m
s$^{-1}$).

**3.3 In-situ zooplankton sampling**
In situ mesozooplankton samples were collected with a WP2 net (60 cm diameter mouth
opening, 300 µm mesh, flowmeter mounted in the net frame) towed vertically from 50 m to
the surface in May 2013, and with a Tucker trawl (1 m$^2$ mouth opening, 300 µm mesh with
flowmeter) used to obtain stratified oblique tows in January 2014 and August 2014 (Table 1).
All samples were preserved in a 5% formaldehyde solution. Zooplankton abundances were
standardized to individuals per m$^3$ of filtered seawater. WP2 vertical tows consisted of 5 depth
intervals from surface to 50 m, every 10 m (0-10, 10-20, 20-30, 30-40, 40-50 m).
Stratified Tucker tows considered four depth strata: 0-10 m, 10-20 m, 20-50 m, 50-100
m in the Puyuhuapi Fjord. In Jacaf Channel, the stratified sampling included five depth strata:
0-10 m, 10-20 m, 20-50 m, 50-100 m and 100-150 m. The hauling speed for both nets was
between 2-3 knots. Sampling occurred during a 36-h period every 3 h from January 22-24,
2014 (Puyuhuapi Fjord) and every 5-6 h from August 18-19, 2016 (Jacaf Channel) (Fig. 1, red
dots). At all sites and dates, zooplankter's were identified, sorted into functional groups,
measured (length) and classified into size-classes using a 5 mm length threshold. To
determine the correlation ($R^2$) between the $S_v$ records from the 38 kHz transducer and the
major macrozooplankton groups (Siphonophores, Chaetognaths and Euphausiids), a quadratic
polynomial curve was also applied between these data sets (further details in section 4.3).

**3.4 Tidal harmonic analysis**
The tidal constituents were computed using HOBO U20 water level loggers and the pressure
sensor from ADCP-3 (Table 1-2, Fig. 1). A tidal harmonic analysis was applied to the sea
level time series according to Pawlowicz et al., (2002), which considers the algorithms of
Godin (1972, 1988) and Foreman (1977, 1978). We classified tides by the dominant period of
the observed tide based on the form factor ($F$), defined by the ratio between the sum of the
amplitudes of the two main diurnal constituents (larger lunar declinational, $O_1$ and luni-solar
declinational, $K_1$) and the sum of the amplitudes of the two main semi-diurnal constituents
(principal lunar, $M_2$ and principal solar, $S_2$), $F = (O_1+K_1)/(M_2+S_2)$ (Bearman , 1989; where, F
< 0.25 semi-diurnal, 0.25 < F< 1.5 Mixed semi-diurnal and F > 3.0 diurnal).

## 4. Results

### 4.1 Hydrographic features

Temperature profiles collected in Puyuhuapi Fjord and Jacaf Channel showed similar structure during the winter and summer campaigns (Fig. 2, a-b). The largest temperature gradients were found between the surface and ~70 m depth, ranging from 8.5° C to 17° C. A thin, fresh layer (salinity values varied from 11 to 29 g kg$^{-1}$) was found in the first ~10 m of the water column below which salinity varied little (29 to ~34.2 g kg$^{-1}$), as result of the presence of Modified Sub-Antarctic water (MSAAW, salinity between 31 and 33 g kg$^{-1}$), the Sub-Antarctic Water (SAAW, salinity between 33 and 33.8 g kg$^{-1}$) and the Equatorial Subsurface Water (ESSW, salinity>33.8 g kg$^{-1}$) (Fig. 2, c-d). Hypoxic conditions (dissolved oxygen below 2 mL L$^{-1}$ and ~30 % saturation) were detected in Puyuhuapi Fjord below 100 m depth, with oxygen concentration between 1-2 mL L$^{-1}$ (Fig. 2e). Deep water in Jacaf Channel was more ventilated, with dissolved oxygen values above hypoxic conditions throughout the water column (Fig. 2f). The hypoxic layer was located over the depth range of the Equatorial Subsurface Water (ESSW) and oxygen rich water (3-6 mL L$^{-1}$) was observed at depths occupied by MSAAW and SAAW. Below 10 m depth, high nitrate concentrations were measured in Puyuhuapi Fjord, but concentrations in the winter (August 2014) were higher than in fall (May 2013) and summer (January 2014) (Fig. 2 g). Along with the in-situ hydrographic sampling, in-situ zooplankton samples were collected and will now be discussed.

### 4.2 ADCP Acoustic data and *in-situ* zooplankton samples

Volume backscatter ($S_v$) from ADCP-1 (50 m depth, May 2013) showed large variability, ranging from high (-90 to -75 dB re 1 m$^{-1}$) to low (-115 to -100 dB re 1 m$^{-1}$) (Fig. 3a). The highest $S_v$ values (>-90 dB re 1 m$^{-1}$) were recorded during the night hours (~18:00 to ~07:00, local time; with all remaining times for in-situ sampling expressed in local time), while minimum $S_v$ values were observed in the daytime (~07:00 to ~18:00) suggesting vertically migrating organisms from deeper waters (below ADCP-1 mooring depth of 50 m) migrate upwards during nighttime hours. From the *in-situ* measurements of macrozooplankton collected at various depth strata in May 2013, the most abundant groups were siphonophores, chaetognaths and medusae (Fig. 3c-f). A marked change in vertical distribution and in total abundance of the macrozooplankton groups in the water column was observed from the first

sampling hour (Fig. 3c) to the night sampling time (~18:00 h), revealing the start of the nocturnal migration to the surface (Fig. 3d) coincident with a DVM pattern as seen in the ADCP-1 backscatter data (Fig. 3a-b).

Data from the ADCP-2 mooring (positioned deeper but at the same location as ADCP-1) from January 22-24, 2014 also showed a strong macrozooplankton DVM pattern, which extended down to ~100 m depth (Fig. 4a). During daylight hours (8:00-18:00), dense aggregations were observed between 80-100 m depth, which started to ascend from 18:00 to 21:00, concentrated close to the surface at night, and began to descend at ~06:00. In-situ stratified sampling showed the most abundant macrozooplankton groups were euphausiids, siphonophores, chaetognaths, decapods and medusae (Fig. 4 b-f). Euphausiids and siphonophores showed higher abundance close to surface layer (10-20 m) during night hours (Fig. 4c and Fig. 4f) and at deeper layers during the daytime (Fig. 4d and Fig. 4e). However, euphausiids showed the clearest diel vertical migration with maximum abundance between 10-20 m layer during night hours, and at ~100 m depth during the daytime (Fig. 4c-f). The in-situ zooplankton samples were complemented by echo-sounder measurements collected along the fjord systems during the summertime and the wintertime. These measurements will now be discussed.

**4.3 Echo-sounder data**

**4.3.1 Summertime single-frequency survey**

The volume backscatter during the summer months overall showed DVM of all macrozooplankton species and a downward migration limit of ~100 m depth due to presence of hypoxic conditions below this depth. Summer daytime $S_v$ values along the Puyuhuapi Fjord averaged -89.1 ±7 dB re 1 m$^{-1}$ and ranged between -110 and -77.3 dB re 1 m$^{-1}$, from the mouth to the head of the Puyuhuapi fjord (Fig. 5a). Most biological backscatter was concentrated in the first 100 m of the water column, matching ADCP-2 results, which showed an increase in backscattering towards 100 m depth (Fig. 4a and 5a). Highest daytime NASC values were found around 80 m (above the hypoxic layer), reaching values of 3-3.5 m$^2$ n mi$^2$ (Fig. 5b). Although some backscatter occurred within the hypoxic layer (below ~120 m depth), all dense aggregations were observed above it (Fig. 5e).

Summer nighttime biological backscattering along the Puyuhuapi Fjord (Fig. 5c) showed maximum $S_v$ values near the surface, suggesting an ascending vertical migration of all

biological backscatter. NASC profiles also showed both an increase in maximum abundances
and a shift in the vertical position of the maximum values from 60-80 m during daytime to
40-60 m depth during nighttime (Fig. 5d). Although the water column depth extended to ~300
m, all dense backscatter aggregations were observed above 100 m depth during both day and
night time hours (Fig. 5a and c). As DO concentrations decreased from 2 mL $L^{-1}$ to 1 mL $L^{-1}$
below 100 m depth, biological scatterers in Puyuhuapi Fjord appeared to prefer oxygen
concentrations between 3 and 7 mL $L^{-1}$ (Fig. 5e). The correlation between $S_v$ values and the
observed density of different zooplankton groups (*in-situ* samples, >5mm) was high. Such
correlations reached values of $R^2$=0.50, for siphonophores (Fig. 6a), $R^2$=0.48 for chaetognaths
(Fig. 6b), and $R^2$=0.72 for euphausiids (Fig. 6c). The wintertime sampling showed similar
findings but was able to capture more activity in the water column due to the use of two
acoustic frequencies.

**4.3.2 Wintertime dual-frequency surveys**
Wintertime dual-frequency surveys data, carried out along Puyuhuapi Fjord and Jacaf
Channel on August 17[th] (~35 km total transect length, Fig. 1), allowed separation of total
backscatter into Fish, Fluid like (FL) and Blue noise (BN) groups (Fig. 7a-b). Total
backscatter ($S_v$) in Puyuhuapi Fjord (0-18 km) showed elevated values in the first 100 m of
the water column, but at slightly deeper depths (50-100 m) than in summer (Fig. 5), possibly
due to bad weather conditions encountered on the sampling day. Greater intensity (-80 to -60
dB re 1 $m^{-1}$) and vertical distribution range (0-220 m) of biological backscattering values
($S_v$>-110 dB) were observed in Jacaf Channel, particularly around its sill (between km 18 and
32; Fig. 7). Particularly high intensities were attributed to BN and FL groups at either side of
Jacaf Channel sill on both August 17[th] and 18[th] (Fig. 7 and 8). An important degree of vertical
segregation between BN and FL groups was also observed along Jacaf Channel, with the first
group concentrated between 100 and 140 m, while the second was between 120 and 200 m
(Fig. 7 and 8).
Continuous acoustic sampling repeated over the Jacaf Channel sill confirmed the
presence of two backscattering layers: one denser layer between 100-150 m and a second, less
dense layer from 200 to 250 m (Fig. 8a, showed only the best echogram). *In-situ* zooplankton
sampling along the Jacaf Channel sill (Fig. 9f) allowed the detection of the major
macrozooplankton groups (e.g., chaetognaths, euphausiids and crustaceans) found during this
experiment (Fig. 9a-d). In general, all sampling stations were carried out during daytime, but
station 4 coincided with the ascending moment of macrozooplankotn, and highlighted the
presence of euphausiids during this time of vertical migration (Fig. 9d). Also, station 1
showed the dominance of crustaceans in the 0-10 m strata. Overall the *in-situ* zooplankton
sampling and the echograms showed good agreement with the FL group (Fig. 9a-d).
Furthermore, the elevated abundance of macrozooplankton groups (euphausiids and
chaetognaths) found between 100-150 m depth during daytime hours (Fig. 9b-f) matched well
with acoustic data for the fluid-like group (Fig. 8a), but in the case of BN group the
macrozooplankton species were not clearly identified in the *in-situ* zooplankton sampling.

A moderate correlation was found between $S_v$ values from Jacaf Channel and

zooplankton density calculated from *in situ* samples (>5 mm), with $R^2=0.42$ for $S_v$ vs.
chaetognaths (Fig. 6d) and $R^2=0.41$ for $S_v$ vs. euphausiids (Fig. 6e). Now the relationships
between water column properties such as temperature, salinity and DO will be compared to
the acoustic and in-situ macrozooplankton measurements.

**4.4 Relationships between biological scattering and water column properties**
To examine relationships between the distribution of biological scattering and water column
properties, $S_v$ values quantified from the 38 kHz acoustic profiler were matched to the
consecutive time at which CTD and DO data were captured. This was done in Puyuhuapi
Channel and Jacaf Channel during the summer and winter seasons, respectively. The
relationship between water temperature and $S_v$ was weak during summer ($R^2=0.30$) and winter
($R^2=0.41$), with maximum $S_v$ values occurring between 8 and 10°C (Fig. 10a and 10b). A
weak relationship was found between $S_v$ and salinity in Puyuhuapi Fjord ($R^2=0.29$, Fig. 10c)
and Jacaf Channel ($R^2=0.35$, Fig. 10d), with higher $S_v$ values found in the MSAAW and
SAAW water masses (salinity >31 g/kg). Both in Puyuhuapi Fjord and Jacaf Channel $S_v$ with
DO and oxygen saturation showed the highest $R^2$ values ($R^2 \sim 0.6$, Fig. 10e-h). Hence, only
20.4% of total $S_v>-110$ dB re 1 m$^{-1}$ were in the hypoxic layer of Puyuhuapi Fjord, while just
1.2 % were in the hypoxic layer in Jacaf Channel (Fig. 10e-h). Now the TKE dissipation will
be discussed to relate macrozooplankton assemblages to vertical mixing in the water column.



## 4.5 Tidal regime

The harmonic analysis carried out with the sea level time series obtained in Puyuhuapi Fjord and Jacaf Channel, denoted the dominance (in terms of amplitude) of the semi-diurnal constituents ($M_2$ and $S_2$; Table 2). Diurnal constituents ($O_1$ and $K_1$) were also important, specifically at the Jacaf ADCP-3 station located close to the Jacaf sill region (Table 2 and Fig 1). The contribution of diurnal constituents added the mixed character to the tidal regimen in the study area. The spectral analysis implemented at all sea level stations showed maximum energy in the semi-diurnal band (Table 2), with the highest spectral energy (57.29 m$^2$ cph$^{-1}$) at Jacaf sill (Jacaf ADCP-3 station), which could be due to the extreme convergence of the channel at this location accelerating the tidal flows.

## 4.6 Mixing process

Turbulence measurements collected with the VMP-250 microstructure profiler showed high dissipation rates of turbulent kinetic energy ($\varepsilon$) in the upper 20 m of the water column in Puyuhuapi Fjord and Jacaf Channel (Fig. 11). In this layer $\varepsilon$ ranged from $10^{-7}$ to $10^{-5}$ W kg$^{-1}$. However, below this surface layer (<20 m depth) the highest values were obtained around Jacaf sill ($\varepsilon=1.2\times10^{-7}$ W kg$^{-1}$), as shown on 21 November 2013 at 140 m depth (Fig. 11 a). In Puyuhuapi Fjord TKE dissipation between 20-180 m was weak ($10^{-10}$ to $10^{-7}$ W kg$^{-1}$), (Fig. 11c and 11e). The dissipation rates of turbulent kinetic energy are obtained by integrating the velocity shear spectrum at each respective depth bin up to the noise limit. The noise limit is determined by comparing the measured spectra to the theoretical Naysmyth Spectra and determining where the measurements begin to deviate from theory. To display how the estimates of $\varepsilon$ were obtained at the Jacaf sill depth, the shear spectra are shown for VMP profiles collected at the Jacaf sill region (21 November 2013 at 140 m depth; Fig. 11b), and in Puyuhuapi Fjord on 22 November 2013 (at 140 m depth; Fig. 11d) and on 23 January 2014 (at 140 m depth; Fig. 11f).

In Puyuhuapi Fjord the correlation between $\varepsilon$ and zooplankton $S_v$ data (38 kHz, fixed station, January 2014) was high ($R^2$=0.65, Fig. 12a). In the same campaign, the *in-situ* macrozooplankton density (>5 mm) was also high correlated with $\varepsilon$ values ($R^2$=0.79 for $\varepsilon$ vs. siphonophores, $R^2$=0.66 for $\varepsilon$ vs. chaetognaths, and $R^2$=0.77 for $\varepsilon$ vs. euphausiids) (Fig 12b-d). Unfortunately, VMP data was not collected in Jacaf Channel in wintertime. In order to confirm the relationship between $\varepsilon$ and various zooplankton species, additional turbulence

measurements were collected in November 2013 along Jacaf sill (Fig. 13a). Results showed
strong velocity shear in the horizontal velocities (Fig. 13b) accompanied by high $\varepsilon$ values ($10^{-7}$
to $10^{-5}$ W kg$^{-1}$; Fig. 11c). Maximum $\varepsilon$ was measured at the Jacaf-Puyuhuapi confluence (10
km along transect) at ~63 m depth where $\varepsilon = 1.9 \times 10^{-5}$ W kg$^{-1}$, (Fig. 13b; St. 164). The
diapycnal eddy diffusivity ($K_\rho$) was also high in the same area with values of $10^{-4}$ to $10^{-3}$ m$^2$ s$^{-1}$
(Fig. 13c).

**5 Discussion**
This study represents one of the first attempts to combine measurements of acoustics,
stratified plankton sampling, microstructure profiles, and standard hydrographic profiles to
investigate both the vertical distribution patterns of macrozooplankton and why these patterns
exist in northwest Patagonian Fjords and other subantarctic latitudes. Three main findings
resulted from this effort. First, DVM patterns of macrozooplankton became evident from all
methodological approaches, at all study periods: May 2013, January 2014 and August 2014
(Fig. 3-5 and Fig. 7-9). Second, strong evidence arose showing macrozooplankton avoidance
of hypoxic layers. And, third, a clear increment of macrozooplankton and fish aggregations
around the Jacaf sill could be related to increased turbulence in this area.

**5.1 Diel vertical migration patterns**
Consistent evidence from multiple echo-sounder surveys, ADCP moorings and semi-
continuous *in-situ* zooplankton measurements supported the existence of major circadian
displacements of macrozooplankton during night hours between mid-depth (20-120 m) and
subsurface waters in our study area. Similar DVM patterns have been found in Reloncaví
Fjord (41.5° S), from 300 and 600 kHz ADCP data, by Valle-Levinson et al., (2014) and
Días-Astudillo et al., (2017) using a 75 kHz acoustic device. Given a greater resolution, the
later work was able to confirm that the DVM affected the whole water column of the fjord
(~200 m). These studies found the presence of euphausiids, decapods, mesopelagic shrimps,
copepods and other groups in the Reloncaví Fjord in July and November, 2006 (Valle-
Levinson et al., 2014), as well as in July 2013 (Días-Astudillo et al., 2017). DVM is a
common feature of many zooplankton groups, observed around the world using different
ADCP and echo-sounders frequencies, e.g., at the Kattegat Channel (Buchholz et al., 1995),
the northeast Atlantic (Heywood, 1996), the northwest coast of Baja California, Mexico
(Robinson and Gómez-Gutiérrez, 1998), the northeastern Gulf of Mexico (Ressler, 2002), the
Antarctic Peninsula (Zhou and Dorland 2004), the Arabian Sea (Fielding et al., 2004), Funka
Bay, Japan (Lee et al., 2004), south Georgia, in the Atlantic sector of Southern Ocean
(Brierley et al., 2006) and Saanish Inlet, Britisch Columbia, Canada (Sato et al., 2013). The
scattering layers observed in these studies highlight the abundances of the major zooplankton
species, represented by: amphipods, euphausiids, siphonophores, chaetognaths, pteropods,
crustaceans, small fish and gelatinous plankton. While most DVM patterns reported in these
studies occurred between 0 and ~300 m depth, the deepest DVM patters were observed in the
North-Atlantic Ocean, reaching depths ~1600 m (Van Haren and Compton, 2013).

DVM patterns of zooplankton are expected to be associated with diel changes in

visible light within the photic zone (from surface to ~100 m). Thus, the zooplankton can avoid
predators during daytime hours and have safe-feeding conditions at night. While only small
irradiance levels, $<10^{-7}$ times surface levels, can be detected beyond 600 m (Van Haren and
Compton, 2013; Sato et al., 2013 and 2016), zooplankton DVM can reach depths below 500
m (Van Haren and Compton, 2013). Moreover, zooplankton DVM occurs in Arctic fjords
(e.g., the Kongsfjorden and Rijpfjorden fjords) even during the polar night, suggesting high
sensitivity to very low levels of solar and/or lunar light (Berge et al., 2009). Since both
Puyuhuapi Fjord and Jacaf Channel are not deeper than 300 m, enough light should reach the
bottom layer and stimulate zooplankton DVM across the whole water column. However, our
results show that zooplankton DVM (and distribution as discussed in the next section) was
limited by the hypoxic boundary layer present in the Puyuhuapi Channel (~100 m; Fig. 5),
providing indirect support to the idea that hypoxia may limit DVM in low-ventilated
Patagonian fjords and elsewhere (Ekau et al., 2010; Mass et al., 2014; Hauss et al., 2016;
Seibel et al., 2016).

**5.2 Macrozooplankton avoidance of hypoxic waters**
In Puyuhuapi Fjord, hypoxic conditions have been reported below ~100 m depth, all year
round (Schneider et al., 2014; Silva and Vargas 2014), with sporadic deep ventilation events
that increase the DO concentration from 1.4 to 2.8 mL $L^{-1}$ (Pérez-Santos, 2017). These
pervasive hypoxic conditions are not common in all Patagonian Fjords. For instance, seasonal
hydrographic data from Reloncaví Fjord showed well ventilated conditions along the fjord,
with deep, near-bottom DO values between 3-3.5 mL $L^{-1}$ (Castillo et al., 2016).
In the current study, acoustic measurements revealed that most biological
backscattering ($S_v$ data) occurred above the hypoxic boundary layer (Fig. 5 and Fig. 10),
which acted as a barrier to DVM and  macrozooplankton distribution throughout the year.
Similar findings were reported in Oslofjord, Norway, where hypoxic conditions dominated
the water column beneath ~60 m depth, and no fish or krill were observed below this depth
(Røstad and Kaartvedt, 2013). Moreover, in Eastern South Pacific OMZ, it has been
previously reported that a number of copepod species and life-stages avoid hypoxic waters
(Castro et al. 1993, Escribano et al. 2009), as well as for most gelatinous zooplankton groups
(Pages et al. 2001; Giesecke and Gonzalez 2005; Escribano et al. 2009). In the same OMZ
region, but further north in Peruvian waters, two diurnal scattering layers were observed, one
over the OMZ and other, mainly composed by adults euphausiids, in the core of the OMZ
(Ballón et al., 2011). Euphausiids, salps and myctophid fish were also observed in the core of
Eastern Tropical North Pacific OMZ (Mass et al., 2014). Seibel et al., (2016) reported
*Euphausia eximia* and *Nematoscelis gracilis* tolerance to hypoxic water and suggest this
tolerance would enable these species to reduce their energy expenditure in at least 50% during
their daytime migration.
The highest $S_v$ values observed in Puyuhuapi Fjord, occurred at DO concentrations
between 2 and 5 mL L$^{-1}$, while in Jacaf Channel between 3 to 6 mL L$^{-1}$. DO values of 3.5 mL
L$^{-1}$ and 4.5 mL L$^{-1}$ seemed to represent appropriate conditions for most macrozooplankton
species in Puyuhuapi Fjord and Jacaf Channel (Fig. 10), respectively, which are similar to the
values indicated by Ekau et al. (2010) for zooplankton. Our results also showed that
macrozooplankton preferred oceanic waters with salinity values >31 g/kg, and temperatures
between 8 and 10° C (Fig. 4, Fig. 9 and Fig. 10). Nonetheless, it must be considered that these
preference values were estimated from observational data and limited sampling rather than
from controlled experiments.
Vertical overlapping observed between fish and macrozooplankton abundances
suggests that the prey-predator interactions might be enhanced under hypoxic conditions.
Pollution and climate change are continually expanding the extent of hypoxic waters around
the world, both in coastal waters and open oceans (Breitburg et al., 2018). While the links
between recent anthropogenic perturbations, such as the salmon aquaculture expansion, and
hypoxia in the Patagonian Fjords is still under debate, it is important to keep this potential
impact upon habitat reductions and enhanced prey-predator interactions under consideration
as it might cause changes in zooplankton groups' distributions and abundance, particularly
those that do not tolerate low DO concentrations.

The fact that some biological backscattering occurred within the hypoxic layer in our

study indicates that hypoxia does not affect all macrozooplankton species equally and that
some of them can inhabit this deeper layer, e.g., euphausiids species (Mass et al., 2014; Seibel
et al., 2016). Hypoxia tolerant species residing below and within minimum DO layers have
been reported, in fact, further north along the Chilean coast during the upwelling season,
leading to support hypotheses on predation evasion and horizontal transport aimed to explain
such behavior (Castro et al., 2007). Within this context *Euphausia pacifica* has been reported
to exhibit the highest abundance of zooplankton species present in hypoxic waters in Hood
Canal, Washington (Sato et al., 2016). Other euphausiids have also been reported to be
present in other hypoxic systems in Chile (Escribano et al, 2009; Gonzalez et al., 2016). It has
been shown that *Euphasia vallentini* is a dominant euphausiid species known to carry out
extensive vertical migrations in Patagonian fjords, hence we speculate it might be one of the
species occurring in the less oxygenated waters of our study. Unfortunately, due to sampling
gear restrictions, we were unable to sample the hypoxic layer, nor to identify firmly the
species occurring at this depth. Therefore, future research will be necessary to understand the
relationship of the deep, yet scarce, macrozooplankton within the hypoxic waters in
Puyuhuapi Fjord. As vertical mixing is a mechanism that could reduce the presence of
hypoxic zones in fjords, values of TKE dissipation were compared to the depth strata of
macrozooplankton.

**5.3 Turbulent mixing at the fjord sill**
Patagonian fjords and channels cover an area of ~240,000 km$^2$ and feature a complex marine
topography, including submarine sills and channel constrictions (Pantoja et al., 2014; Inall
and Gillibrand, 2010). Bernoulli aspiration, internal hydraulic jumps and intense tidal mixing
are all processes that can be found near a fjord sill (Farmer and Freeland, 1983; Klymark and
Gregg, 2003; Inall and Gillibrand, 2010; Whitney et al., 2014). Our data showed elevated
values of TKE dissipation in Jacaf Channel ($\varepsilon =10^{-5}$ W kg$^{-1}$ and $K_\rho =10^{-3}$ m$^2$ s$^{-1}$) near the sill
from 0-60 m depth. These values are similar to those observed at the sill of Knight Inlet in
Canada (Klymark and Gregg, 2003). Lower $\varepsilon$ values were found in Puyuhuapi Fjord (Fig.11).
The elevated vertical mixing (high $K_\rho$) in Jacaf Channel is probably due to the barotropic tide
interacting with the submarine sill (Schneider et al., 2014; Fig. 11, Fig.13 and Table 2). This
was also observed in Martinez Channel (Pérez-Santos et al., 2014), Central Patagonia, where
semidiurnal internal tides were found to dominate the estuarine dynamics (Ross et al., 2014).
This region is highly influenced by the Baker river, whose discharge enhances stratification
and introduces suspended solids that subsequently limit productivity in the water column
(González et al., 2010; Daneri et al., 2012; González et al., 2013).

The evident aggregation of macrozooplankton and fish found near Jacaf sill (within ~1

km) matches the area exhibiting the highest $\varepsilon$ values (~$10^{-5}$ W kg$^{-1}$; Fig.13). Thin (2-5 m) and
thick (10-50 m) regions of enhanced vertical shear measured directly with the VMP-250
microstructure profiler contribute to vertical mixing. Subsequently this enhances the exchange
between the subsurface rich nutrient layer (Fig. 2) and the photic layer, leading to increased
phytoplankton productivity (Montero et al., 2017a; Montero et al., 2017b), as shown in the
conceptual model of figure 14. Thus, the acoustic and turbulence measurements collected near
Jacaf sill promote the importance of a sill in influencing the vertical distribution of oxygen,
macrozooplankton and fish on both sides of the sill.

A summary of the processes that can contribute to macrozooplankton vertical

distribution and aggregation in Puyuhuapi Fjord and Jacaf Channel are presented in a Fig. 14.
In Puyuhuapi Fjord, at 100 m depth a high nutrient and high production layer (Daneri et al.,
2012; Montero et al., 2017a; Montero et al., 2017b) is separated from a hypoxic layer below,
which limits species distribution and lacks significant aggregations of zooplankton. Above the
hypoxic waters, turbulent mixing enhances contact between macrozooplankton predators and
their prey (Visser et al., 2009). In Jacaf Channel, the hypoxic layer occurs deeper in the water
column than in Puyuhuapi Fjord, which stretches the vertical distribution of
macrozooplankton to a deeper range. Turbulent mixing also increases primary and secondary
production, through enhanced nutrient availability and favors encounters of
macrozooplankton with potential prey, increasing growth and survival rates (Visser and Stips
2002; MacCready et al., 2002; Klymak and Gregg 2004; Lee et al., 2005; Visser et al., 2009;
Whitney et al., 2014).

**5.4 Other findings and considerations**

Results showed similar groups of macrozooplankton (>5 mm) in Puyuhuapi Fjord and

Jacaf Channel: euphausiids, chaetognaths, medusae and siphonophores during summer
(January 2014) and winter (winter 2014). However, euphausiids were not observed in fall
2013, which was an unexpected result which deserves further confirmation and analysis. In
contrast, fall 2013 sampling presented the highest acoustic abundances within the time series
(Fig. 3). The elevated accumulation of macrozooplankton species around the sill may impose
a significant modification in the amount and quality of carbon exported to deeper waters in
particular zones of the fjords. Future studies on carbon flux quantification in fjords should
incorporate sill regions to test this hypothesis, in order to improve ocean pumping
assessments in the context of climate change and variability.

**6 Conclusions**
This paper was aimed to determine how hypoxic conditions affect the vertical distribution of
macrozooplankton in fjords and to assess how vertical mixing relates to abundances of
macrozooplankton at fjord sills. Results showed that the hypoxic layer in Patagonian Fjords
limits DVM and overall distribution of macrozooplankton to the upper ~100 m of the water
column, reducing the habitat of these species. The hypoxic zones were found away from
underwater sills or areas that would experience enhanced turbulence. When assessing the
abundance of macrozooplankton in conjunction with TKE dissipation near a submarine sill it
was found that elevated turbulence generated by the barotropic tide interacting with the sharp
bathymetric feature enhanced vertical mixing, deepened the hypoxic layer and injected
nutrients. In addition, macrozooplankton were found in higher densities and extended deeper
in the water column around the submarine sills. This is thought to be due to an increase in
primary production that would result from the effects of elevated vertical mixing.

**Acknowledgment**
The ADCP data was collected as part of the FONDECYT Grant 3120038 and 11140161 by
Dr. Iván Pérez-Santos and the help of Dr. Wolfgang Schneider's research group. We thank
Dr. Arnoldo Valle-Levinson for motivating the acoustic study of zooplankton in Chilean
Patagonia. We also thank Dr. Luis Cubillos and Dr. Billy Ernst for providing the scientific
echo-sounder and Cristian Parra and Hernán Rebolledo for administering the scientific echo-
sounder sampling. COPAS Sur-Austral CONICYT PIA PFB31 financed part of the field work
and Lauren Ross's trips to Patagonia. We thank Juan Ramón Velasquez and Oscar Pizarro
research group for his assistance in the ADCP 1, 2 and 3 moorings and Adolfo Mesa, Aldo
Balba and Eduardo Escalona for conducting most of the zooplankton sampling. Giovanni
Daneri is funded by FONDECYT Grant 1131063.

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

 **Figure captions**

 Figure 1. Study area in relation to South America and the Pacific Ocean is the small panel in

 the top right. The main figure enlarges the study area (Puyuhuapi Fjord and Jacaf Channel)

 and indicates the instruments used for data collection, fixed point station positions, and the sill

 location near the head of Jacaf Channel. The contours indicate the depth of the fjords.

 Figure 2. (Upper panel) Profiles of temperature (a-b), salinity (c-d), dissolved oxygen (e-f)

 and nitrate (g) collected during different oceanographic campaigns in the northern central part

 of Puyuhuapi Fjord and (lower panel) in eastern region of the Jacaf Channel.

 Figure 3. (a) Volume backscattering strength ($S_v$,) calculated from the ADCP-1 backscatter

 signal in Puyuhuapi Fjord, deployed at 50 m depth from the 8$^{th}$ to the 26$^{th}$ of May, 2013. (b)

 Zoom of the $S_v$ data and the times of *in-situ* zooplankton sampling (black dots) carried out

 during May 25-26, 2013. (c-d) Vertical abundance of main zooplankton groups (>5 mm

 length) from the *in-situ* sampling at 16:00 and 18:00 (local time) on May 25$^{th}$ and (e-f) at 9:00

 and 11:00 (local time)  on May 26$^{th}$.

 Figure 4. (a) Volume backscattering strength ($S_v$) calculated from the ADCP-2 backscatter

 signal in Puyuhuapi Fjord from the 22$^{nd}$ to the 24$^{th}$ of January, 2014. The *in-situ* zooplankton

 sampling (in 3:00 intervals) are represented by black dots at the surface. (b) Depth integrated

 abundance of zooplankton from the surface to 100 m depth varying throughout time, where

 the top panel is zooplankton > 5 mm in length.  (c) Vertical abundance of the principal

 zooplanktons groups on January 23$^{rd}$ at 2:00 (night time) and (d, e and f) same as (c) but on

 January 23$^{rd}$ at 08:00 and 14:00 (daytime) and January 24$^{th}$ at 02:00 (night time). The time

 reference is in local time.

 Figure 5. Single frequency (38 kHz) scientific echo-sounder transect conducted along the

 Puyuhuapi Fjord during the Summertime field campaign (January 2014). Distribution

 indicated by colors representing $S_v$. (a) Daytime transect of echo-sounder measurements ($S_v$)

 throughout depth (*y*-axis) from the mouth (0 km) to the head (80 km) of Puyuhuapi Fjord on

 January 22, 2014. (b) Average profiles derived from the Nautical Area Scattering Coefficient

 (NASC) from the daytime transect with standard deviation bars, (c) Same as (a), but for the

 night time starting at 21:57 (local time) January 24th through early in the morning of January

 25, 2014. (d) Same as (b) but for the nighttime. The ADCP-2 mooring location is marked with

 a black dot in (a) and (c). (e) Dissolved oxygen profiles (black dots) obtained approximately

every three hours (close to the position of ADCP-2 mooring) from January 23$^{rd}$ to 24$^{th}$, 2014.
The location of the hypoxic boundary layer is depicted by the white contour line of 2 mL L$^{-1}$.
Figure 6. Scatter plot of volume backscattering strength ($S_v$) from 38 kHz frequency and the
most abundance macrozooplankton species obtained in the *in-situ* fixed stations carried out in
Puyuhuapi Fjord (a, b and c) during January 22-24, 2014 and (d, e) in Jacaf Channel during
August 18-19, 2014.
Figure 7. Dual-frequency (38 and 120 kHz) scientific echo-sounder transects along Puyuhuapi
Fjord (0-18 km) and Jacaf Channel (18-35 km) during nighttime on August 17, 2014. (a)
Fluid like and (b) blue noise echogram for zooplankton and (c) the fish echogram.
Distribution indicated by colors representing *Sv* values. The black arrow in (a) represents the
entrance to Jacaf Channel. Horizontal red lines in (a, b, c) denote lower limits of usable
acoustic data (250 m).
Figure 8. Dual-frequency (38 and 120 kHz) acoustic transect across Jacaf sill conducted
during daytime on August 18, 2014. (a) Fluid-like echogram, (b) blue noise echogram for
zooplankton and (c) the fish echogram. Distribution indicated by colors representing $S_v$
values. Horizontal red lines in (a, b, c) denote lower limits of usable acoustic data (250 m).
Figure 9. (a-d) *In-situ* stratified zooplankton sampling along Jacaf Channel during August 17,
2014 and the acoustic data collected simultaneously using the dual-frequency (38 and 120
kHz). FL is fluid-like and BN is blue noise groups. (e) Depth integrated abundance of
macrozooplankton groups from surface to 150 m depth for various sampling hours. (f)
Showed the stations positions. (g-j) The vertical abundance of the main macrozooplankton
groups found during the wintertime survey.
Figure 10. Relationships between the relative abundance of zooplankton (expressed in $S_v$
values) using 38 kHz frequency echo-sounder measurements (*y*-axis) and temperature in (a)
Puyhuhuapi Fjord and (b) Jacaf Channel; salinity in (c) Puyuhuapi Fjord and (d) Jacaf
Channel; dissolved oxygen in (e) Puyuhuapi Fjord and (f) Jacaf Channel; oxygen saturation in
(g) from Puyuhuapi Fjord and (bh) Jacaf Channel. The black lines denote the quadratic fit
curves, contour colors indicate depth.
Figure 11. Profiles of water temperature (blue line), vertical shear (red line) and dissipation
rate of turbulent kinetic energy (black line with green dots) obtained with the VMP-250
microprofiler at the depth of the Jacaf sill (~140 m depth) in (a) Jacaf Channel on 21
November 2013 (c) Puyuhuapi Fjord on 22 November 2013 and (e) in Puyuhuapi Fjord on 23

January 2014. (b, d, f) Reprentative spectrum of velocity shear ($\partial$u/$\partial$z) for shear probe 1 (blue line) and 2 (red line) in wavenumber space in Jacaf Channel on 21 November 2013, Puyuhuapi Fjord on 22 November 2013 and Puyuhuapi Fjord on 23 January 2014, respectively. The black line denotes the dimensional Nasmyth spectrum and the red and blue triangles the cut-off of maximum wavenumber ($k_{max}$) for each shear probe. The shear spectrums were carried out in the same layer (135-145 m) for all turbulence profilers.

Figure 12. Scatter plot of dissipation rate of turbulent kinetic energy ($\varepsilon$) and (a) volume backscattering strength ($S_v$) from 38 kHz frequency and (b, c and d) the most abundance macrozooplankton species obtained in the *in-situ* fixed stations carried out in Puyuhuapi Fjord during Junuary 22-24, 2014.

Figure 13. (a) Microstructure profile locations along Jacaf Channel and sill using VMP-250 in November 2013. (b) The color bar showed the dissipation rate of turbulent kinetic energy *($\varepsilon$)* and the blue lines depict the velocity shear at each station location along Jacaf Channel (as shown in (a). The horizontal scale (-2 to 2 s$^{-1}$) applied to profiles at stations 160, 162 and 163. Station 164 is located at the confluence of Jacaf Channel and Puyuhuapi Fjord (10.5 km) (c) The diapycnal eddy diffusivity profiles ($K_p$), obtained at each station shown in (a).

Figure 14. Conceptual model to show the oceanographic processes that contribute to the distribution and aggregation of zooplankton in (a) Puyuhuapi Fjord and (b) Jacaf Channel.

**Table captions**

Table 1. Data set collected during oceanographic campaigns in Puyuhuapi Fjord and Jacaf Channel.

Table 2. Harmonic analysis implemented to water level time series in Puyuhuapi Fjord and Jacaf Channel.

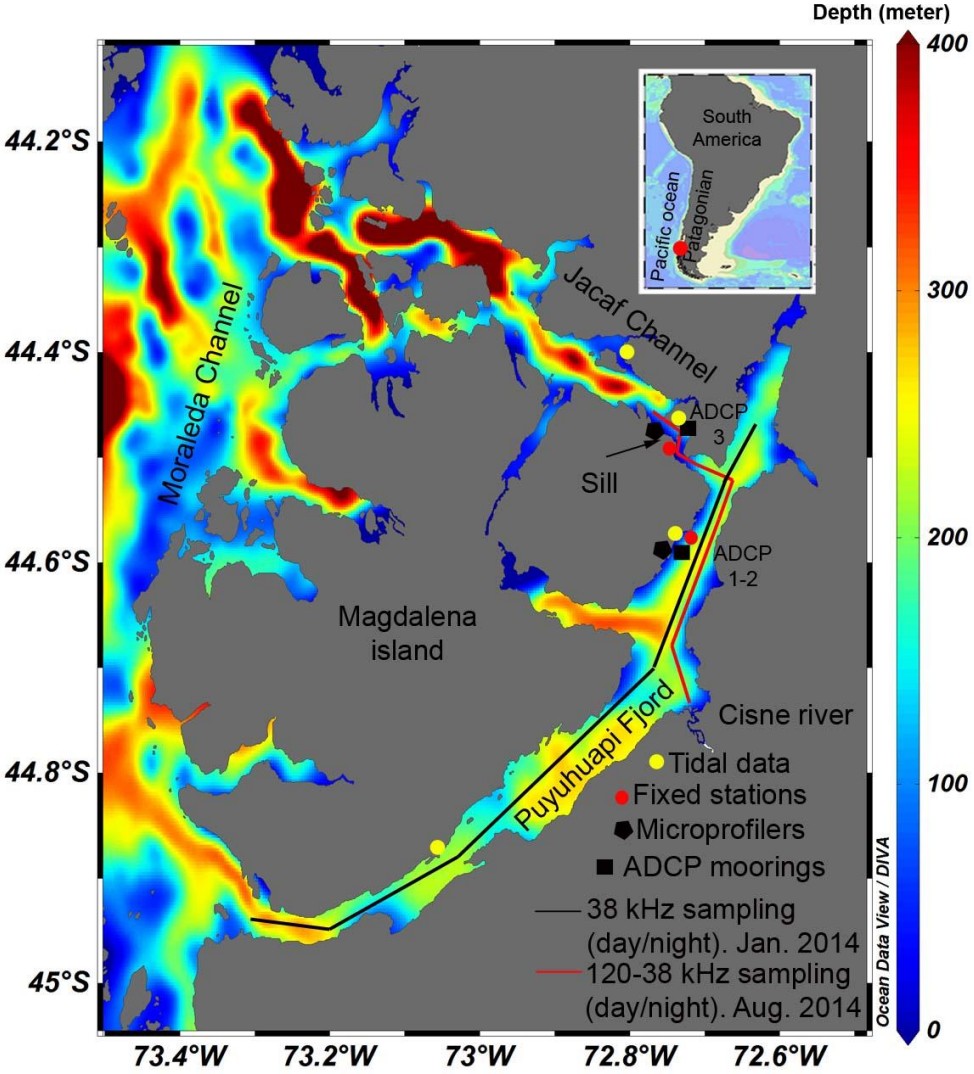

1119

1120

Figure 1. Study area in relation to South America and the Pacific Ocean is the small panel in
the top right. The main figure enlarges the study area (Puyuhuapi Fjord and Jacaf Channel)
and indicates the instruments used for data collection, fixed point station positions, and the sill
location near the head of Jacaf Channel. The contours indicate the depth of the fjords.


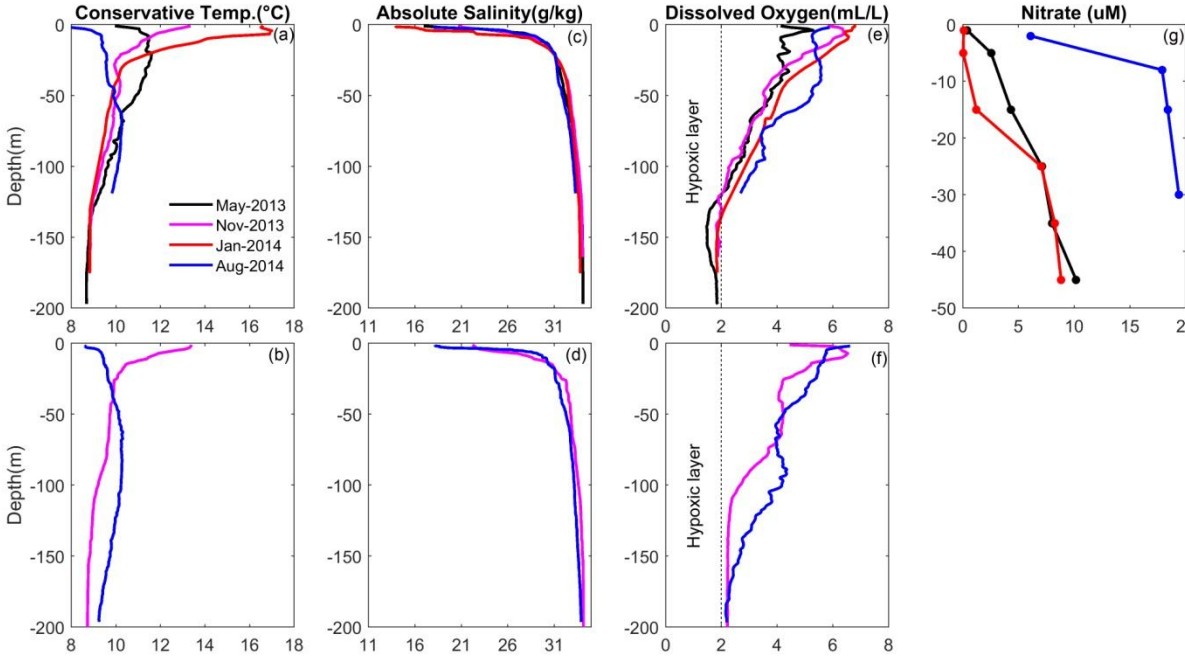


Figure 2. (Upper panel) Profiles of temperature (a-b), salinity (c-d), dissolved oxygen (e-f)
and nitrate (g) collected during different oceanographic campaigns in the northern central part
of Puyuhuapi Fjord and (lower panel) in eastern region of the Jacaf Channel.

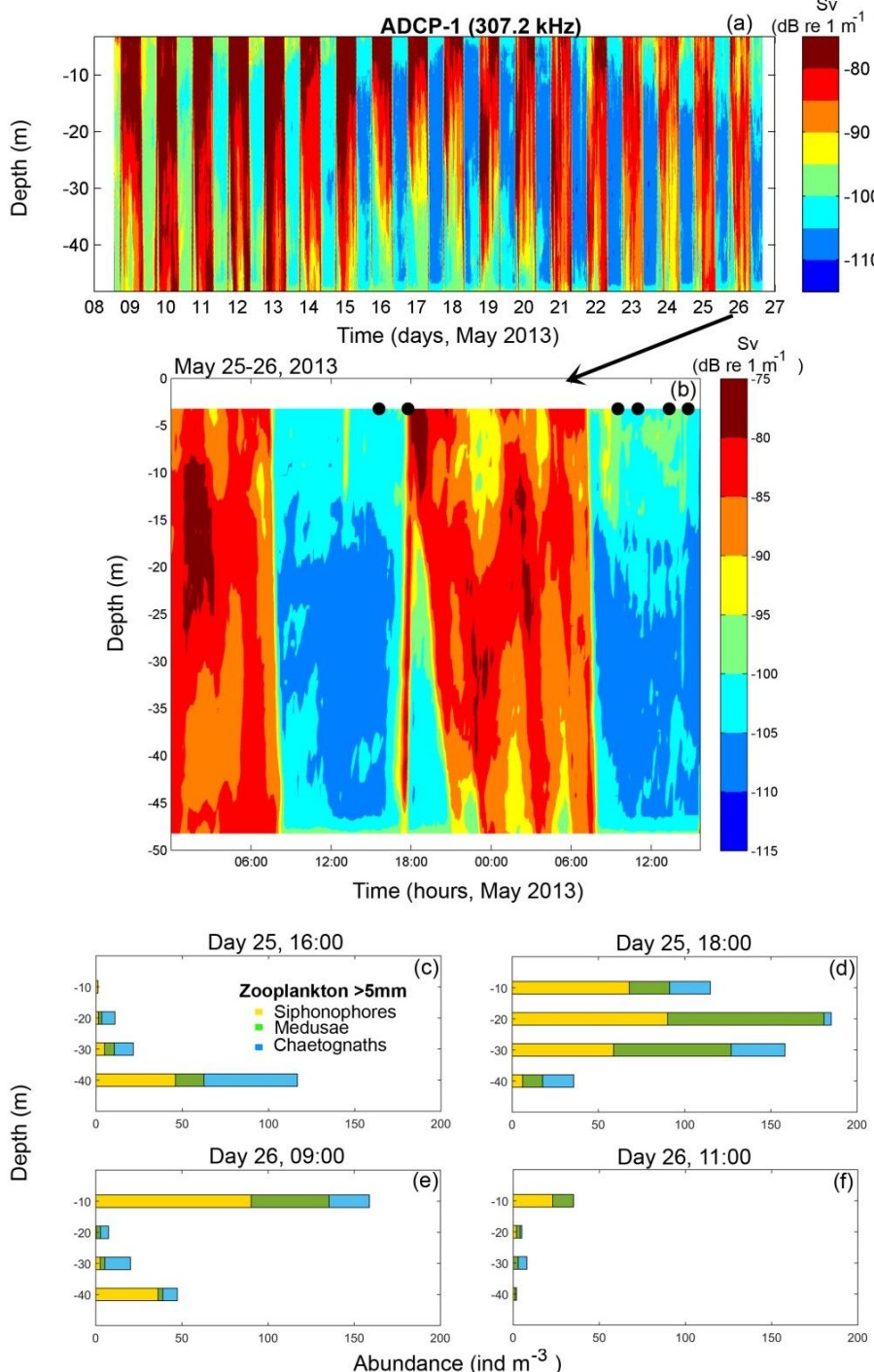


Figure 3. (a) Volume backscattering strength ($S_v$) calculated from the ADCP-1 backscatter

signal in Puyuhuapi Fjord, deployed at 50 m depth from the 8[th] to the 26[th] of May, 2013. (b)

Zoom of the $S_v$ data and the times of *in-situ* zooplankton sampling (black dots) carried out

during May 25-26, 2013. (c-d) Vertical abundance of main zooplankton groups (>5 mm

length) from the *in-situ* sampling at 16:00 and 18:00 (local time) on May 25[th] and (e-f) at 9:00

and 11:00 (local time) on May 26[th].

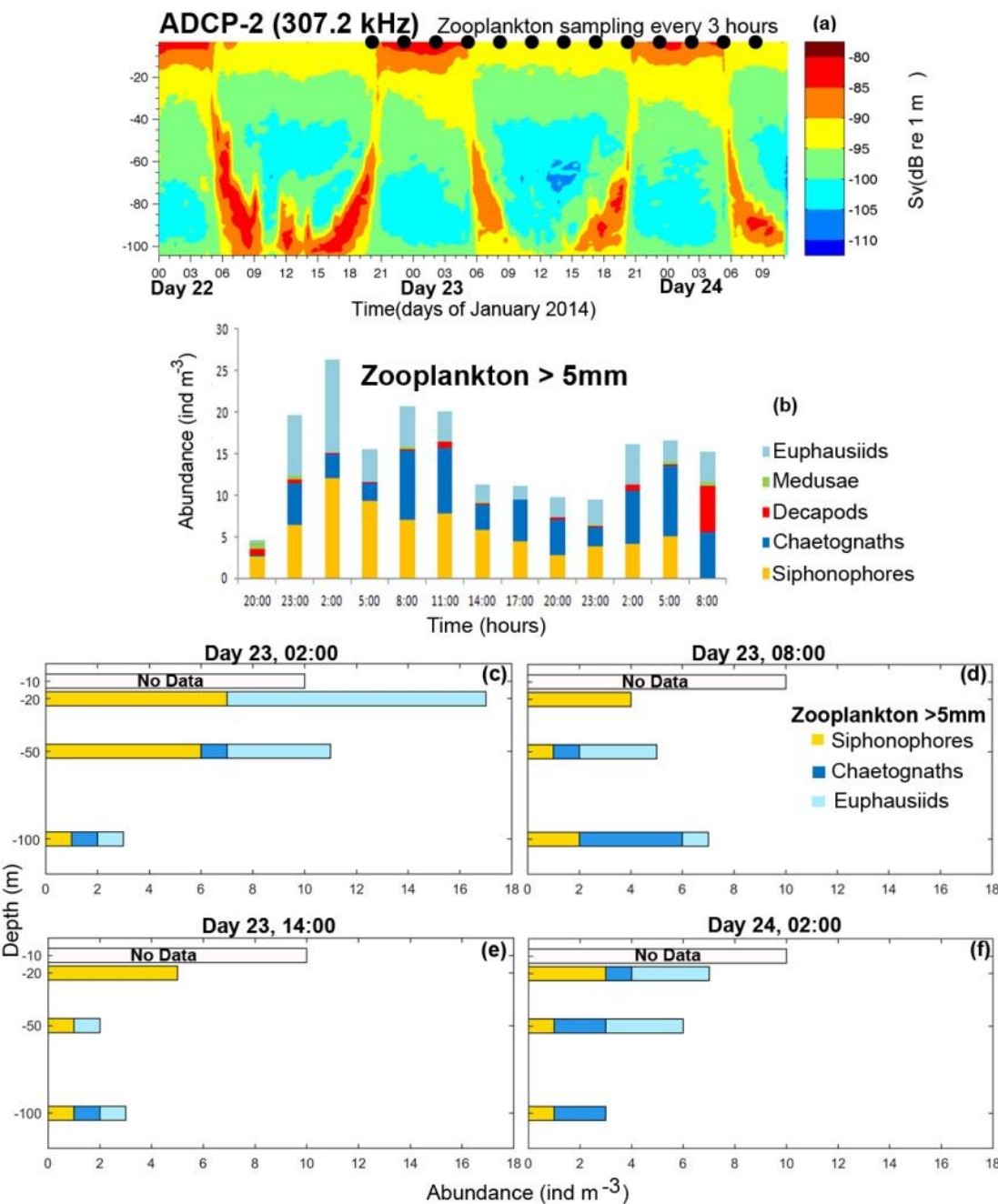

1137

Figure 4. (a) Volume backscattering strength ($S_v$) calculated from the ADCP-2 backscatter
signal in Puyuhuapi Fjord from the 22nd to the 24th of January, 2014. The *in-situ* zooplankton
sampling (in 3:00 intervals) are represented by black dots at the surface. (b) Depth integrated
abundance of zooplankton from the surface to 100 m depth varying throughout time, where
the top panel is zooplankton > 5 mm in. (c) Vertical abundance of the principal zooplanktons
groups on January 23rd at 2:00 (nighttime) and (d, e and f) same as (c) but on January 23rd at
8:00 and 14:00 (daytime) and January 24th at 02:00 (nighttime). The time reference is in local
time.

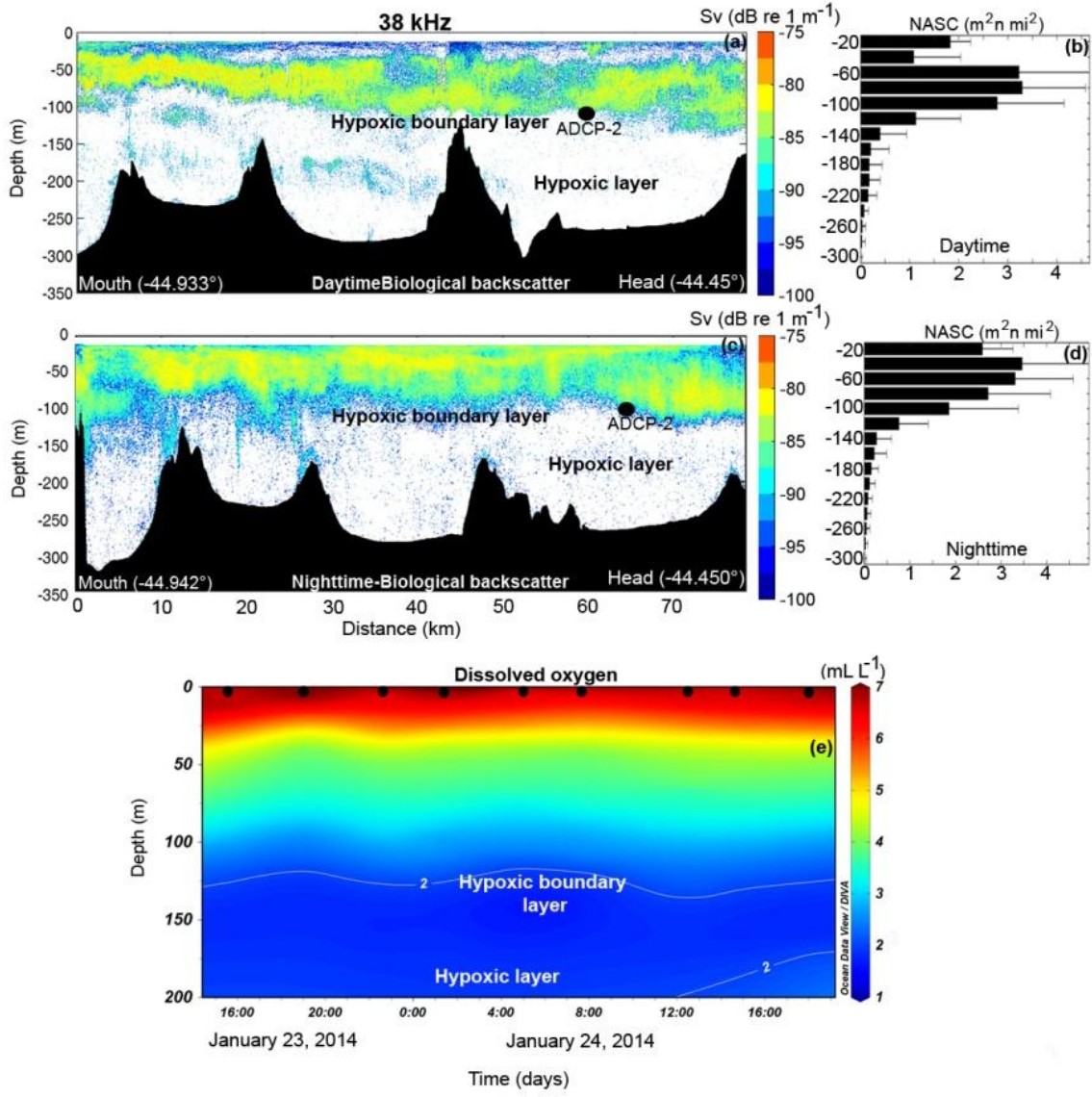

Figure 5. Single frequency (38 kHz) scientific echo-sounder transect conducted along the Puyuhuapi Fjord during the summertime field campaign (January 2014). Distribution indicated by colors representing $S_v$. (a) Daytime transect of echo-sounder measurements ($S_v$) throughout depth (*y*-axis) from the mouth (0 km) to the head (80 km) of Puyuhuapi Fjord on January 22, 2014. (b) Average profiles derived from the Nautical Area Scattering Coefficient (NASC) from the daytime transect with standard deviation bars. (c) Same as (a), but for the nighttime starting at 21:57 (local time) January 24th through early in the morning of January 25, 2014. (d) Same as (b) but for the nighttime. The ADCP-2 mooring location is marked with a black dot in (a) and (c). (e) Dissolved oxygen profiles (black dots) obtained approximately every three hours (close to the position of ADCP-2 mooring) from January 23[rd] to 24[th], 2014. The location of the hypoxic boundary layer is depicted by the white contour line of 2 mL L[-1].

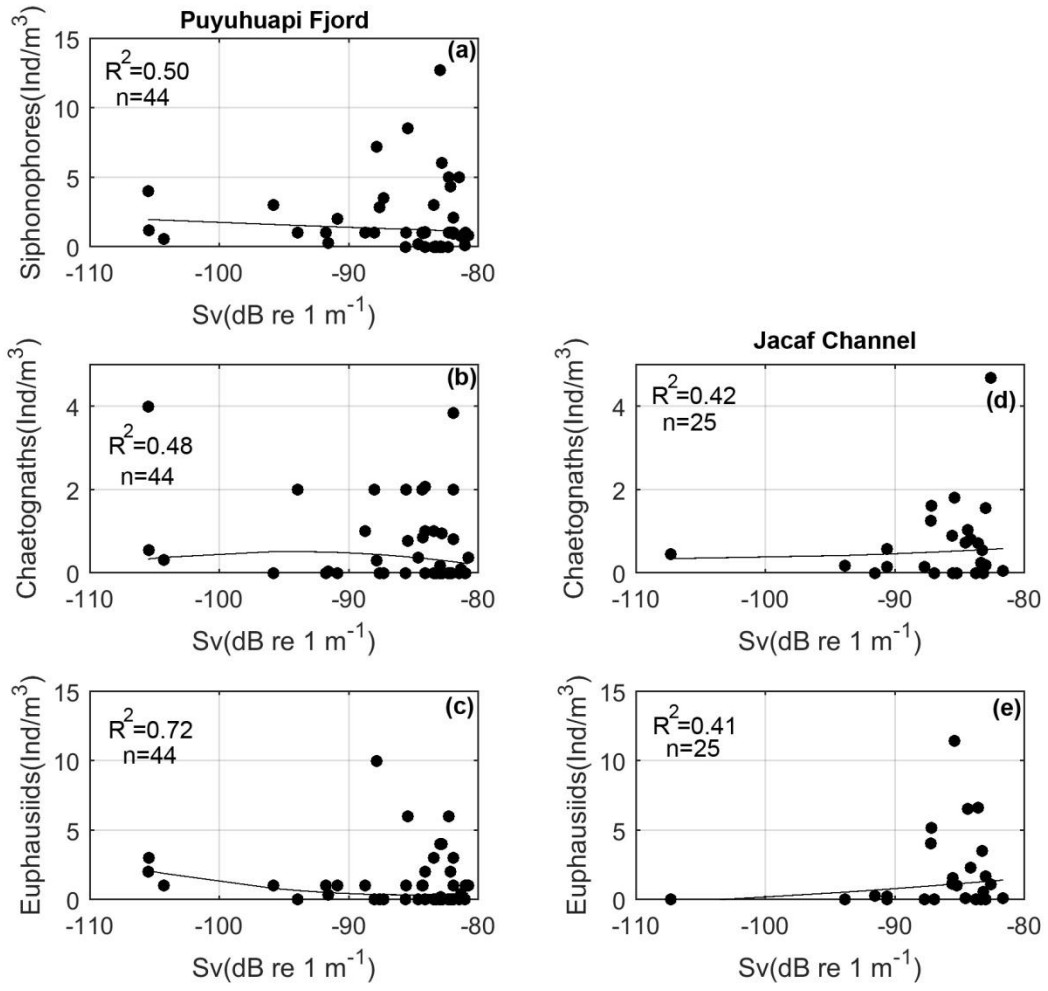


Figure 6. Scatter plot of volume backscattering strength ($S_v$) from 38 kHz frequency and the
most abundance macrozooplankton species obtained in the *in-situ* fixed stations carried out in
Puyuhuapi Fjord (a, b and c) during January 22-24, 2014 and (d, e) in Jacaf Channel during
August 18-19, 2014.




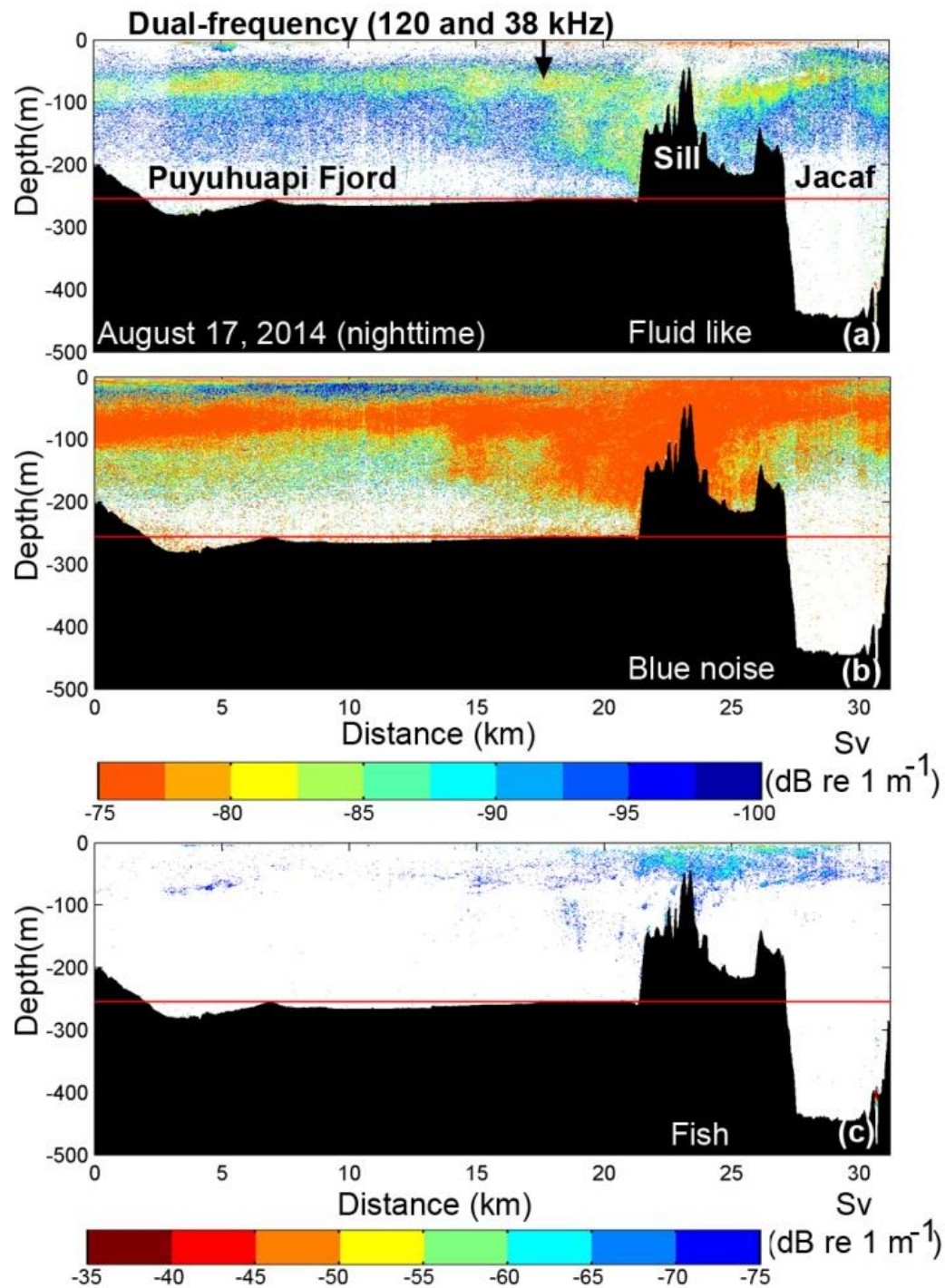

Figure 7. Dual-frequency (38 and 120 kHz) scientific echo-sounder transect along Puyuhuapi Fjord (0-18 km) and Jacaf Channel (18-35 km) during nighttime on August 17, 2014. (a) Fluid like and (b) blue noise echogram for zooplankton and (c) the fish echogram. Distribution indicated by colors representing *Sv* values. The black arrow in (a) represents the entrance to Jacaf Channel. Horizontal red lines in (a, b, c) denote lower limits of usable acoustic data (250 m).

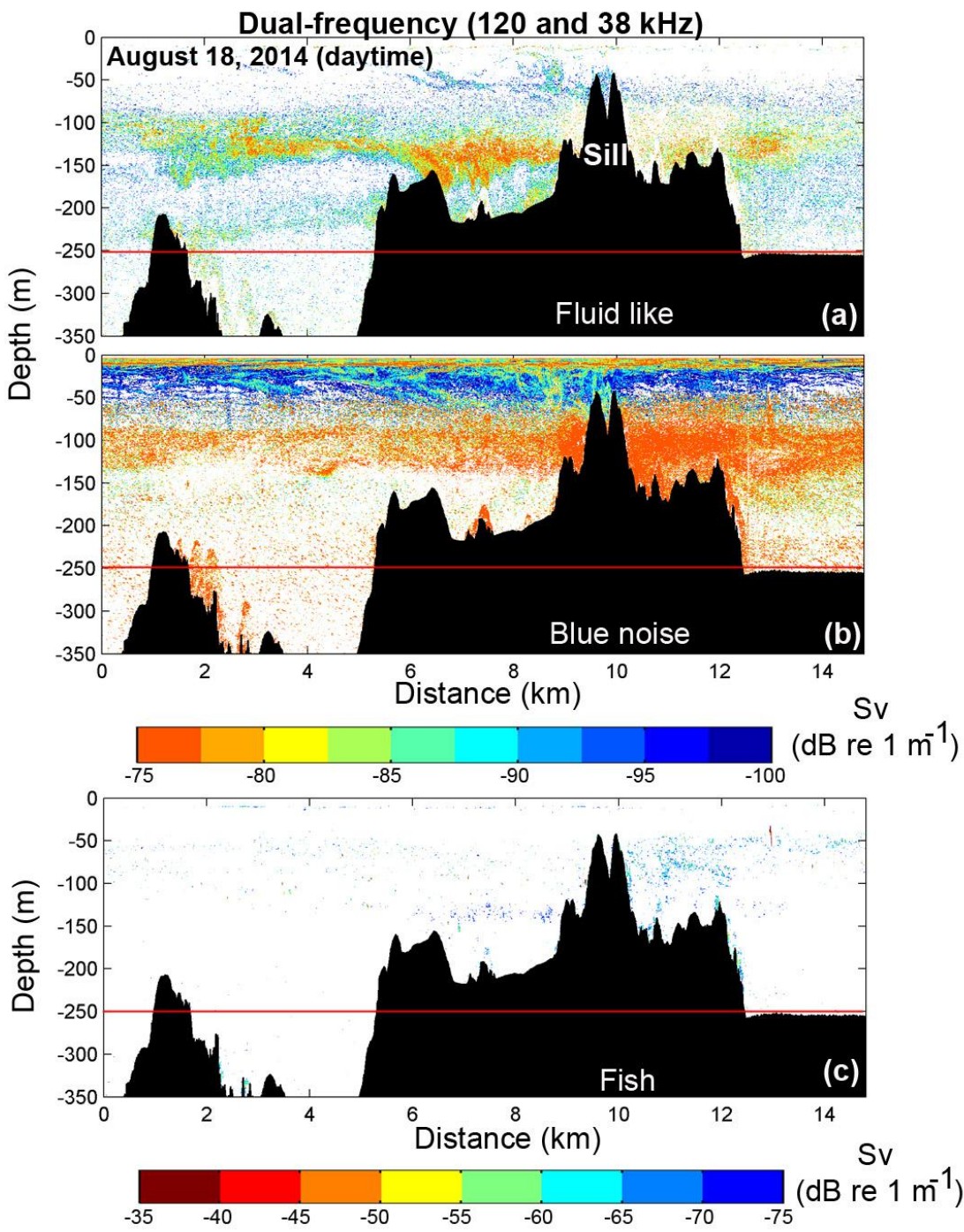

Figure 8. Dual-frequency (38 and 120 kHz) acoustic transect across the Jacaf sill conducted during daytime on August 18, 2014. (a) Fluid-like echogram, (b) blue noise echogram for zooplankton and (c) the fish echogram. Distribution indicated by colors representing $S_v$ values. Horizontal red lines in (a, b, c) denote lower limit of usable acoustic data (250 m).

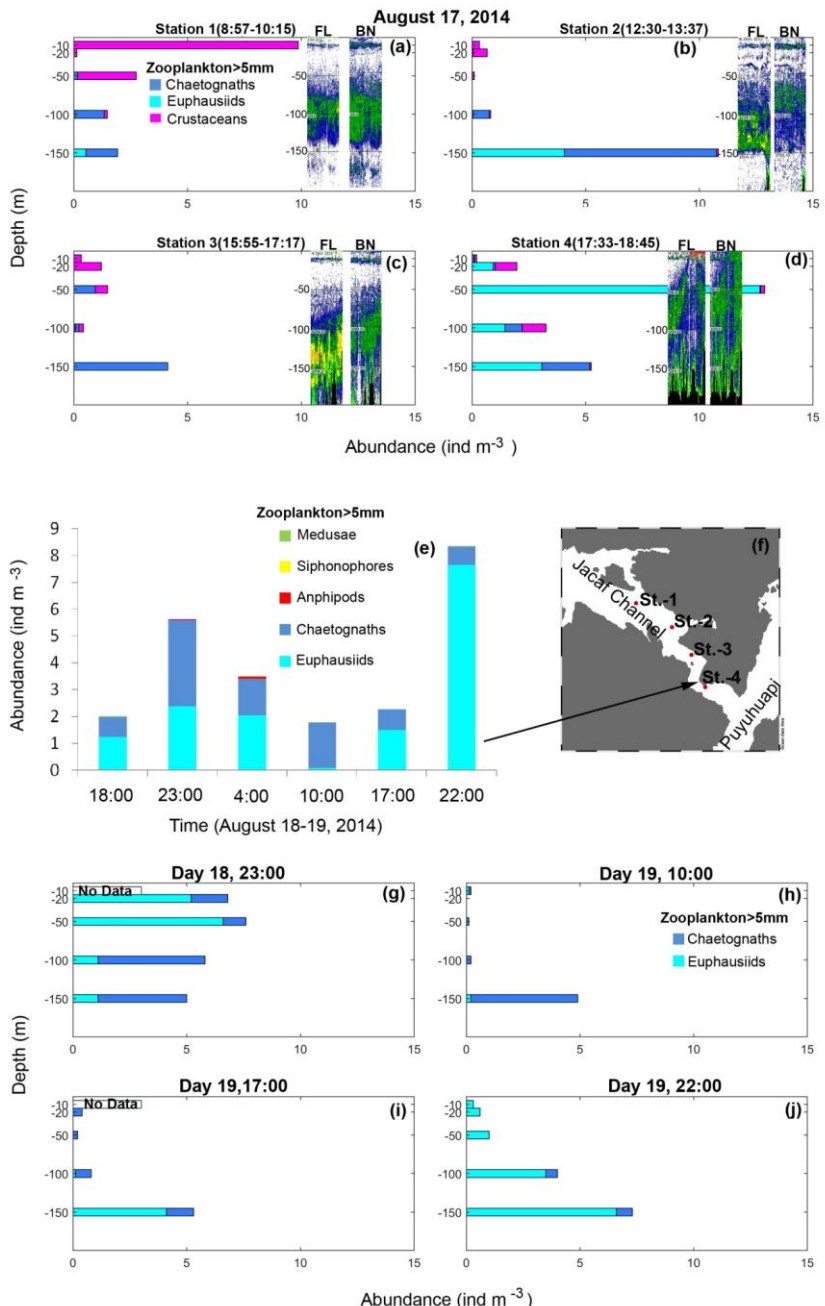

1181

Figure 9. (a-d) *In-situ* stratified zooplankton sampling along Jacaf Channel during August 17, 2014 and the acoustic data collected simultaneously using the dual-frequency (38 and 120 kHz). FL is fluid-like and BN is blue noise groups. (e) Depth integrated abundance of macrozooplankton groups from surface to 150 m depth for various sampling hours. (f) Showed the stations positions. (g-j) The vertical abundance of the main macrozooplankton groups found during the wintertime survey.

1188

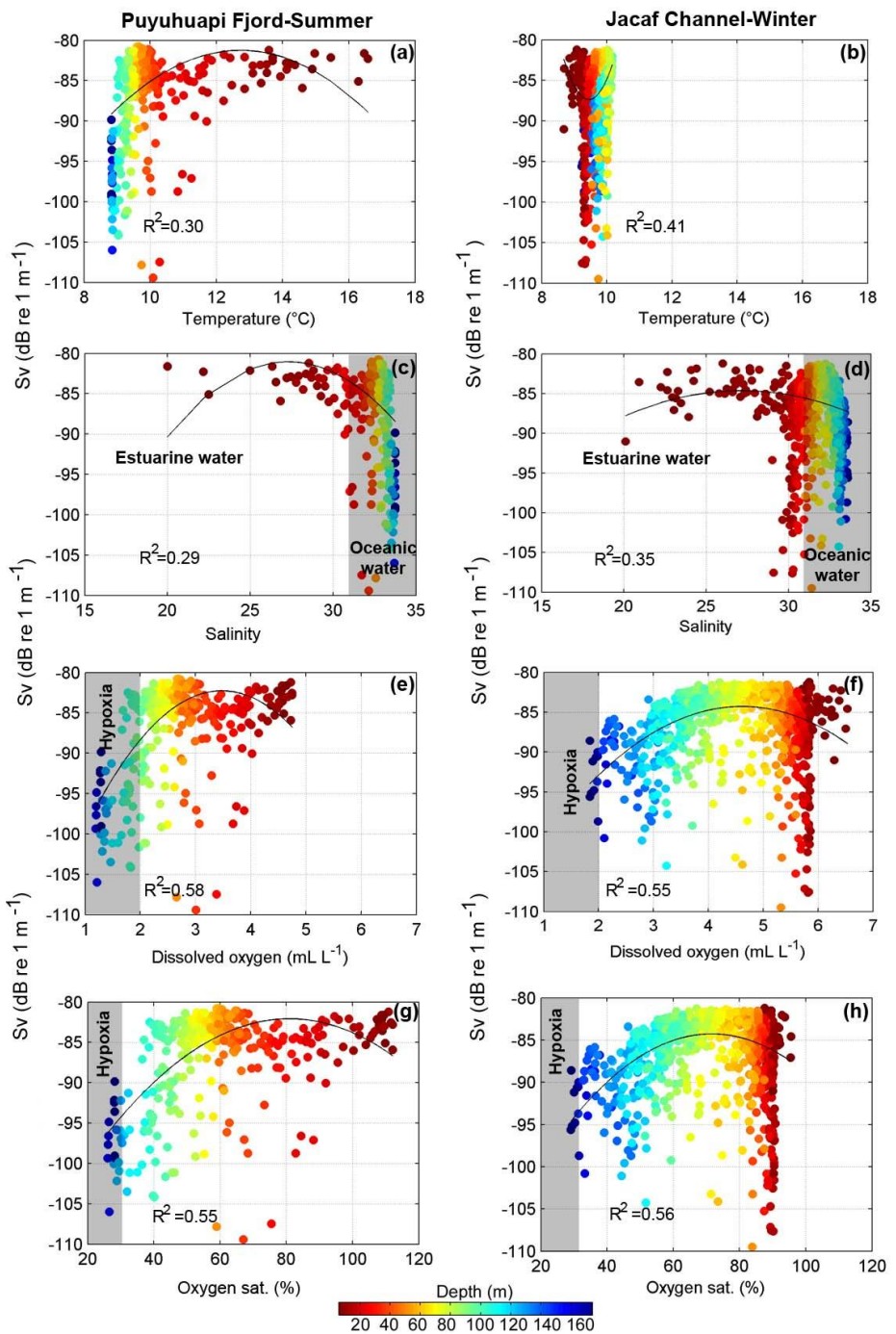

1189

Figure 10. Relationships between the relative abundance of zooplankton (expressed in $S_v$ values) using 38 kHz frequency echo-sounder measurements (*y*-axis) and temperature in (a) Puyhuhuapi Fjord and (b) Jacaf Channel; salinity in (c) Puyuhuapi Fjord and (d) Jacaf Channel; dissolved oxygen in (e) Puyuhuapi Fjord and (f) Jacaf Channel; oxygen saturation in (g) from Puyuhuapi Fjord and (h) Jacaf Channel. The black lines denote the quadratic fit curves, contour colors indicate depth.

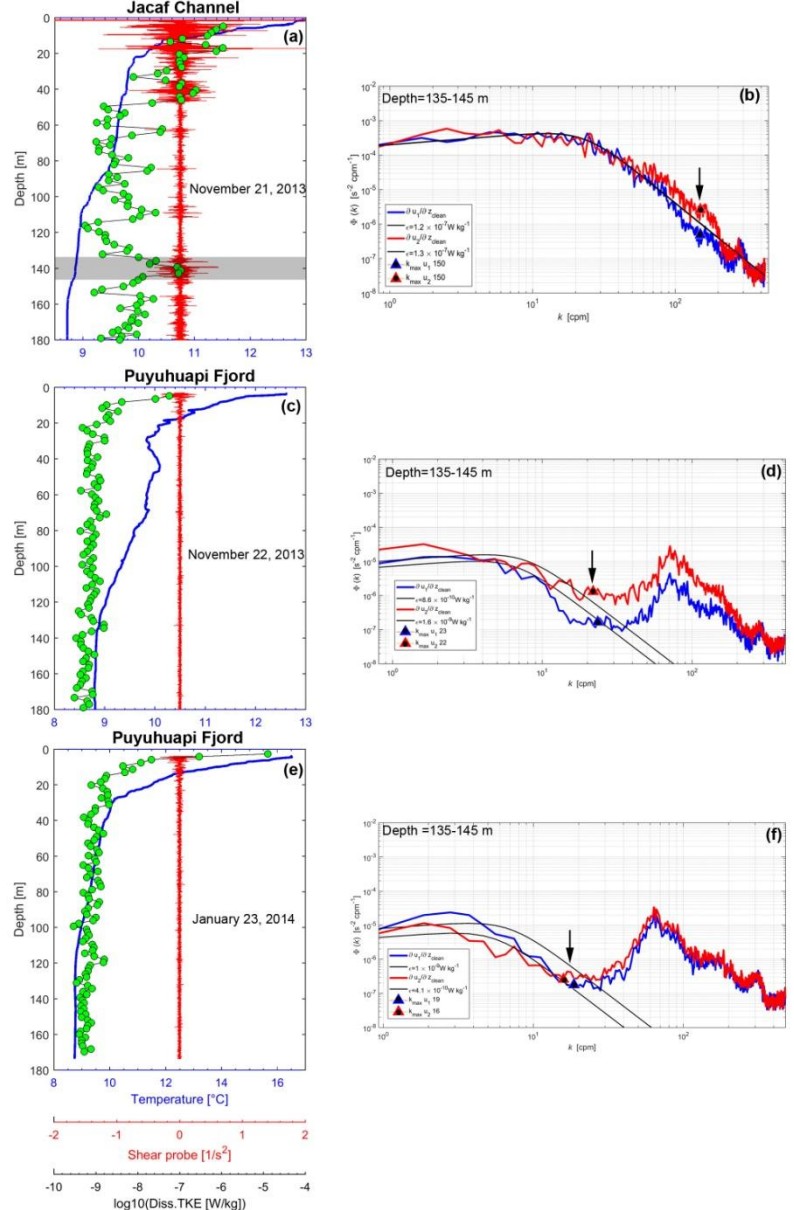

1196

Figure 11. Profiles of water temperature (blue line), vertical shear (red line) and dissipation rate of turbulent kinetic energy (black line with green dots) obtained with the VMP-250 microprofiler at the depth of the Jacaf sill (~140 m depth) in (a) Jacaf Channel on 21 November 2013 (c) Puyuhuapi Fjord on 22 November 2013 and (e) in Puyuhuapi Fjord on 23 January 2014. (b, d, f) Reprentative spectrum of velocity shear ($\partial u/\partial z$) for shear probe 1 (blue line) and 2 (red line) in wavenumber space in Jacaf Channel on 21 November 2013, Puyuhuapi Fjord on 22 November 2013 and Puyuhuapi Fjord on 23 January 2014, respectively. The black line denotes the dimensional Nasmyth spectrum and the red and blue triangles the cut-off of maximum wavenumber ($k_{max}$) for each shear probe. The shear spectrums were carried out in the same layer (135-145 m) for all turbulence profilers.

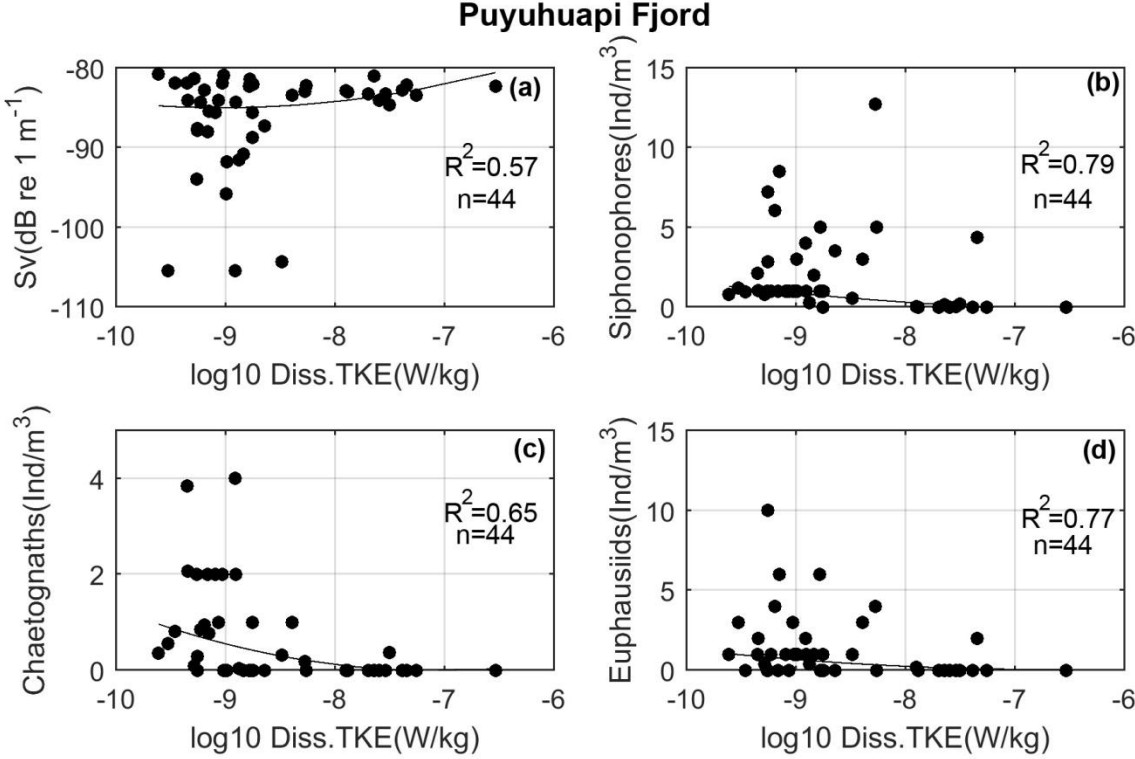


Figure 12. Scatter plot of dissipation rate of turbulent kinetic energy (ε) and (a) volume
backscattering strength ($S_v$) from 38 kHz frequency and (b, c and d) the most abundance
macrozooplankton species obtained in the *in-situ* fixed stations carried out in Puyuhuapi Fjord
during January 22-24, 2014.



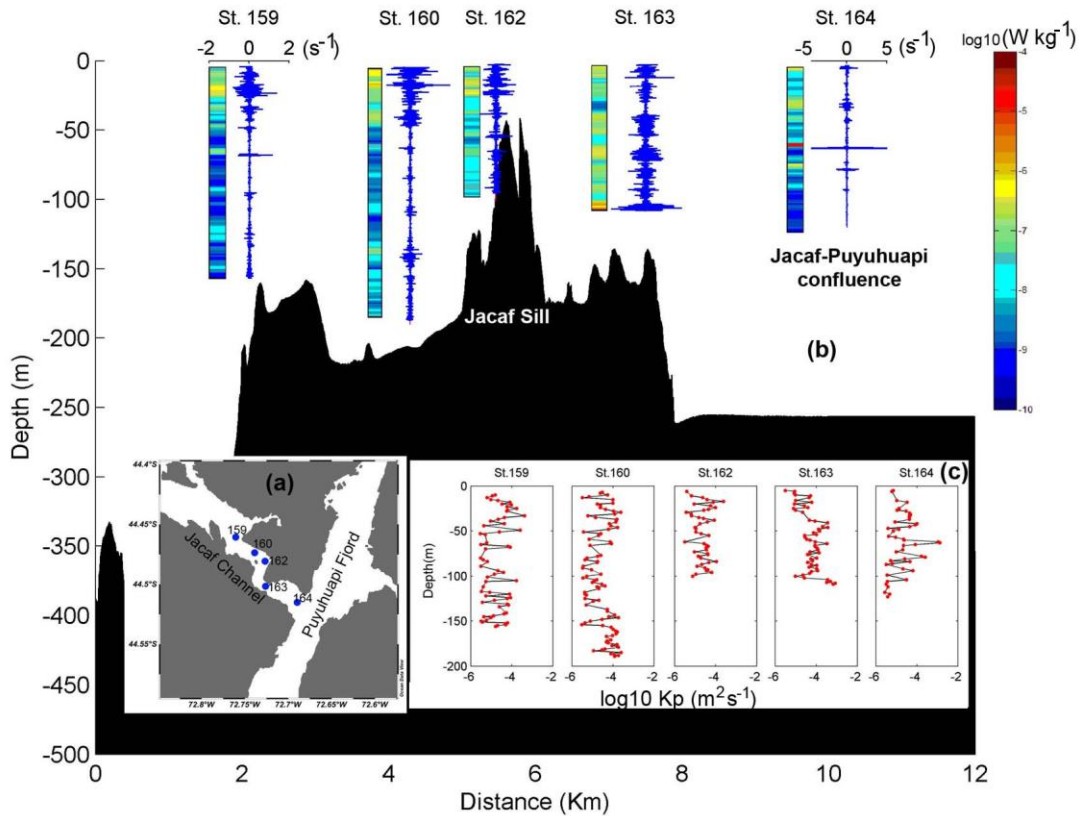


Figure 13. (a) Microstructure profile locations along Jacaf Channel and sill using VMP-250 in
November 2013. (b) The color bar showed the dissipation rate of turbulent kinetic energy *(ε)*
and the blue lines depict the velocity shear at each station location along Jacaf Channel (as
shown in (a). The horizontal scale (-2 to 2 s$^{-1}$) applied to profiles at stations 160, 162 and 163.
Station 164 is located at the confluence of Jacaf Channel and Puyuhuapi Fjord (10.5 km) (c)
The diapycnal eddy diffusivity profiles ($K_p$), obtained at each station shown in (a).


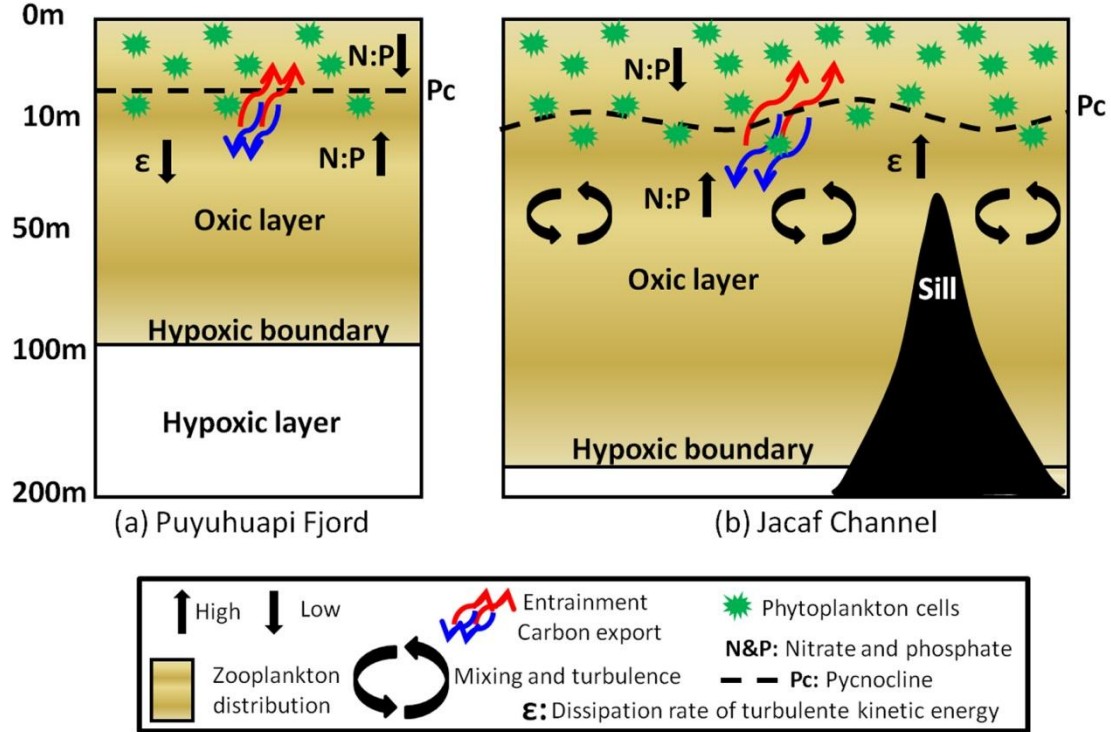


Figure 14. Conceptual model to show the oceanographic processes that contribute to the
distribution and aggregation of zooplankton in (a) Puyuhuapi Fjord and (b) Jacaf Channel.

Table 1. Data set collected during oceanographic campaigns in Puyuhuapi Fjord and Jacaf
Channel.

| Location | Date | Season | Data measured | Instruments |
|---|---|---|---|---|
| **Puyuhuapi Fjord** | May 8-27, 2013 | Fall | -Acoustic data 307.7 kHz | -ADCP-1 RDI |
| | | | -Zooplankton | -WP2 net |
| | | | -Hydrography | -CTD SBE-25 |
| | | | -Nitrate | Spectrophotometry |
| | November 22, 2013 | Spring | -Turbulence | -VMP-250 |
| | | | -Hydrography | -CTD SBE-25 |
| | January 22-25, 2014 | Summer | -Acoustic data 307.7 kHz | -ADCP-2 RDI |
| | | | -Acoustic data 38 kHz | -SIMRAD EK60 |
| | | | -Zooplankton | -Tucker Trawl net |
| | | | -Turbulence | -VMP-250 |
| | | | -Hydrography | -YSI 6600 |
| | | | -Nitrate | Spectrophotometry |
| | August 17-19, 2014 | Winter | -Acoustic data 38 and 120 kHz | -SIMRAD EK60 |
| | | | -Zooplankton | -Tucker Trawl net |
| | | | -Hydrography | -CTD SBE-25 |

| | February-June 2016 | Summer-Fall | -Nitrate -Tidal data (south) | Spectrophotometry -HOBO U20 |
| | February-November 2016 | Summer-Spring | -Tidal data (north) | -HOBO U20 |
| | June 16, 2016 | Fall | -Hydrography | -CTD SBE-25 |
| **Jacaf Channel** | April-November 2012 | Fall-Spring | -Tidal data | -HOBO U20 |
| | November 21, 2013 | Spring | -Turbulence | -VMP-250 |
| | August 2014-May 2015 | Winter-Spring–Summer-Fall | -Hydrography -Tidal data | -CTD SBE-25 -ADCP-3 |
| | August 17-19, 2014 | Winter | -Acoustic data 38 and 120 kHz -Zooplankton -Hydrography | -SIMRAD EK-60 -Tucker Trawl net -CTD SBE-25 |


Table 2. Harmonic analysis implemented to water level time series in Puyuhuapi Fjord and
Jacaf Channel.

| Sea level time series | Date (mm-yyyy) | Energy from semi-diurnal band ($m^2$ $cph^{-1}$) | Amplitude of principal constituents (cm) | | | | F | Tidal regime |
|---|---|---|---|---|---|---|---|---|
| | | | $M_2$ | $S_2$ | $O_1$ | $K_1$ | | |
| Jacaf-HOBO | 04-09/ 2012 | 45.10 | 83.45 | 28.32 | 14.46 | 22.33 | 0.32 | Mixed semi-diurnal |
| Jacaf-ADCP | 08/2014-05/2015 | 57.29 | 60.67 | 61.01 | 57.78 | 42.48 | 0.82 | Mixed semi-diurnal |
| Puyuhuapi-HOBO south | 02-06/2016 | 44.45 | 81.97 | 31.51 | 13.37 | 18.36 | 0.27 | Mixed semi-diurnal |
| Puyuhuapi-HOBO north | 02-11/2016 | 49.17 | 89.15 | 31.07 | 11.03 | 17.75 | 0.23 | Semi-diurnal |
