# Peer review of "Turbulence and hypoxia contribute to dense biological scattering layers in"

_Ocean Science, 2017_

## Referee Comment (RC1) · Anonymous Referee #1 · 29 Nov 2017

**Summary**

Perez-Santos et al. use a dataset comprised of hydroacoustics coupled with hydrographic measurements and zooplankton samplings to examine the effects of physical and chemical properties on zooplankton distributions in a Patagonian Fjord. Interdisciplinary data collected in this study is remarkable, covering biological, physical, and chemical properties. The authors achieved extensive coverage both temporally and spatially by combining moored and ship-based surveys. However, this manuscript lacks clear objectives and significance of the study. As a result, I cannot comment on the significance of the study. The Introduction contains specific information about the

Patagonian fjords, which are more relevant to the section titled "study area", without placing the study to the larger context. In addition, the paper does not do an adequate job analyzing the echosounder data. While the paper focuses on the zooplankton distribution, frequency of echosounder chosen (i.e., 38 kHz) is not relevant to examine zooplankton. Although the observations may be of interest, this manuscript is not yet ready for publication.

**General comments**

Abstract does not contain the objectives of this study, instead heavily focused on the methods. There is fairly detailed description about the study site, which is more appropriate to place in the main text.

Introduction should include the objectives and significance of the study. Currently, there is not enough description on knowledge gap in this field based on previous studies and justification of the study site. Detailed description of the study site is more appropriate to place in the Methods.

I have major concerns on the analysis of echosounder data to extract zooplankton backscatter. In general, zooplankton species found in their net samples (e.g., copepods, euphausiids) cannot be detected at 38 kHz, because they are too small to be significant backscatterers compared to the wavelength of 38 kHz. How did the authors separate zooplankton from fish (Fig. 6)?

When they have two frequencies (following Ballon et al. 2011), Sv data from 120 kHz is useful for up to  $\sim$  200 m depth due to the increase in background noise with range. However, the data analysis was conducted up to  $\sim$  450 m depth (Fig. 7). Only data within the analysis limit should be examined.

There is no discussion on the seasonal change in zooplankton distributions and compositions. Based on their seasonal coverage of the data sets, seasonal component should be considered in addition the difference in study sites. The manuscript needs to be carefully reviewed for typos and grammatical errors. Comments from co-authors (lines 710-712) remained in the main text, which need to be removed. Section numbers are not in sequence in the Results, Discussion, and Conclusions.

Technical comments

Description of the data collection is complicated and hard to follow because there are many sensors deployed during different times of the year at different locations. Inclusion of a table summarizing the details (e.g., types of data collected, deployment locations/depth, period of data collection) would increase the readability of the manuscript.

The method lacks detailed description of the echosounders, such as ping rate, calibration information, and preprocessing of the data (e.g., bottom detection, near-field removal, background noise removal).

Use of the word, echosounders, should be consistent throughout the text. The authors use "echo sounders", "echo-sounders", and "echosounders", which need to be fixed.

Sv units, dB re 1 m-1, should be used throughout the text. The authors often use "dB" toward the end of the manuscript and figure legends and captions.

Fig. 1: Color should be consistent between two colorbars. In the manuscript, red is SHALLOWER depth in the overview map, while red is DEEPER depth in zoomin figure, which are confusing. Some symbols overlap each other, which makes the readers difficult to understand the legends.

Fig. 2: Content of the figures on the top row overlaps with Fig. 3, and the patterns of the Jacaf Channel are very similar to those in the Puyuhuapi Fjord. Also, data points from previous studies are not discussed in the text. Thus, this figure could be removed from the manuscript.

Fig. 3: Define "MSAAW", "SAAW", and "ESSW" in the figure caption. No definition of MSAAW and ESSW is stated in the text either. X-axis should be distance from the

mouth, instead of latitude, because the fjord is positioned diagonally.

Fig. 4: What does "AFIOBIOEX" mean? This term is not introduced in the text, but only appears in the figure captions (Fig. 5 as well). To improve the readability of the manuscript, AFIOBIOEX should be removed from the captions.

Fig. 5: The bars showing the standard deviation are not legible in (c)-(e). There is no x- and y-labels.

Fig. 6: X-axis of (a, c), and (d, f) is not consistent. All figures should be corrected for distance from the same reference point (e.g., distance from the mouth). What do the numbers in (b) and (e) mean? The upper bound of the hypoxia layer needs to be included, because it is not clear where the hypoxic layer is located.

Fig. 7: There is no need to plot the same data at two different frequencies. 38 kHz for fish and 120 kHz for zooplankton are commonly used in bioacoustic field.

Fig. 9: Which frequency is used for Sv values?

Below is some examples of typos: Remove a period from the title.

Line 69: change "has" to "have".

Line 98: add comma after "advection".

Line 104: add comma after "CTD profiles".

Line 128-129: "northern mouth" cannot be identified in Fig. 1, because the subset of the figure blocks the portion of the fjord map.

Lines 143: Be consistent for the use of numbers (e.g., one vs. 1).

Line 205: What does "CITA" mean?

Line 210: Ballon et al. (2011) is not in the References.

Line 210: Unit of NASC is "m2 nmi-2". The equation of NASC does not need to be

presented, because this is a common knowledge.

Lines 240-241: Remove the references, because these are commonly used techniques.

Lines 243-247, 277-279: There is no need to include the study plan that did not happen.

Line 266: What does "ESSW" mean?

Line 274: Change "<" to ">".

Lines 336-337: There is no time on the x-axis of Fig. 6. Time should be included on the x-axis, so that the readers can follow your interpretation.

Line 371: Change "+" to "and". "DO" should be defined and used throughout the text, instead of using both DO and dissolved oxygen.

Line 414: "Others" should be "other".

Lines 425-428: This sentence is contradicting. Did you mean there is twilight vertical migration, or not?

Line 469: Remove "(Fig. 10)". This is duplication.

Line 480: Cut "in" before "there might be".

---

## Referee Comment (RC2) · Anonymous Referee #2 · 18 Dec 2017

The manuscript aims at relating the vertical distribution and migration of zooplankton to physical structures and turbulence in Chilean fjord system. This is a timely and interesting focus. However, I find the manuscript in its present form preliminary and of local interest only. The difficulties are: The objectives of the study appear primarily of technical nature. The relevance and implications of studying the vertical distribution in relation to fine scale properties and turbulent mixing needs to be highlighted in more detail. The introduction and the discussion basically lack scientific questions related to the physical-biological interactions and do not relate to an already large body of literature about detecting zooplankton with acoustic methods or the influence of physical (turbulence) or chemical (oxygen-minimum zones) properties on zooplankton distribution. The implied effects on reproduction, growth and life cycles in the introduction are not sufficient and appear redundant because the physical and biological processes occur on very different time cycles. Reference to previous work is largely restricted Chilean fjords.

In addition, the material and methods are incomplete and inconsistent. Many details can be found below. It is unclear to me, why hydrographical data from 1995-2015 is presented, while zooplankton sampling is restricted to a few occasions. Data on zooplankton from net sampling in August 2014 is not presented although samples were apparently taken; instead physical data from 2016 is presented although not described in the Methods.

Finally, the authors make very little use of their own data, particularly with regard to the identification of the primary groups responsible for the detected backscattering signals. The zooplankton depth resolved data should be presented and analysed. Data from 2013 suggests that copepods contribute very little to the signal, but the authors treat the backscatter data as equivalent to zooplankton throughout the manuscript. Tremendous differences in the abundance of zooplankton despite similar backscatter signal strength needs to be explained.

In its present form, I cannot recommend considering the manuscript for publication and suggest that the authors revise it considerably.

Detailed comments:

Introduction Line 70: Palma (2008) is missing the reference list Line 74: Landaeta et al. (2013) is missing in the reference list. When microzooplankton and fish larvae were studied, copepods (meso- and macrozooplankton) cannot dominate. Line 80 following: Rephrase the sentence. Why 'although'? What is meant by accurate results? Nets and acoustic methods provide principally different results with high taxonomic resolution in the first and high spatial resolution in the second. Thus, they are used to study different aspects and differ largely in their size resolution. Line 88: Please specify:
Norwegian Channel or Kattegat? Line 89: Buchholz et al. 1995, Zhou and Dorland 2004 are missing in the reference list. Line 89 following: The necessity and need for studying the vertical distribution or migration in relation to physical properties needs to be described better. They are themselves not a scientific question. Line 97: Please specify the implications for reproduction and growth. Yamasaki et al. 2002 is missing in reference list. Line 101: The influence of the described processes (short-term) on biological life cycles (different time scales) needs to be explained. Line 108: What is meant by 'survival strategies present in these organisms'? Line 111: It is unclear to me what the stage-specific migration patterns of Rhincalanus have to do with the effect of fine scale turbulence patterns. Acoustics cannot be used to resolve the stages of this species. Line 115: The introduction lacks a review of the present knowledge about the effect on turbulent mixing and the oxygen conditions on the distribution of zooplankton. What are the scientific questions? What zooplankton is targeted at? Nets and acoustic profilers provide largely different type of data.

Material Methods Line 146: Specify the depths for nutrient samples. Line 201: The description of the echo sounder needs to be checked. On line 188, it says SIMRAD CX 34 at 38 kHz, here it says EK-60 at 38 and 120 kHz. Please specify also how the echo intensity was combined Line 210: Please explain the units (what is n and mi<square>. Zooplankton abundance is usually presented per unit volume, thus the full units should be presented (also of T) Line 217: What is 'Tx' in the formula? Line 238: The sampling in 2013 covered only the upper 50 m but not the watercoulmn of 100 m scanned by the ADCP. Why? Line 249: No information is presented on the analysis of the sampling. From the data presented in the study no differentiation into size classes was performed. Why?

Results: Line 254 following: In Fig 2, the top 100 m should be resolved because the size of the graphs make it very difficult to extract the information on T, S, and the other variables. The legend should be self-explanatory, but it is not. The T at the surface is 15 degrees, the x-axis stops at 14 degrees. The text and figures do not always
match: during Puy V hypoxic water occurred at a depth > 200m and not as implied by the text at >100 m. Line 263: The MatMeth indicate that the sampling covered the period 1995-2015. Now data from 2016 are presented. This is confusing. Why was this data included? Line 275: Sampling was conducted in layers of 10 m depth, but data on zooplankton distribution is averaged. Why? Information on size classes should be presented. In addition: was the abundance integrated as indicated in the figure legend? Then m-3 is wrong. Fig. 4 c does not allow extracting quantitative information on siphonophores. Figure 5: Apparently, zooplankton was analysed in size categories; this needs to be described in the methods. A lot of information is lost by averaging/integration (this is not clear to me; it looks like averaging but integration is stated). I suggest to present the zooplankton data (size, taxa) as in Figure 5a despite a courser resolution. Then, signal and zooplankton distribution can be compared. Negative abundances in Fig c-e are odd. Zooplankton abundance in Jan 2014 (daytime) is several orders of magnitude lower than in 2013 (daytime), but signal strength appears similar or even higher. This needs explanation. Line 303: The authors describe here that copepods and others together contribute to the signal in backscatter. This is not conclusive until the data is shown in high resolution as described above. In addition, a similar analysis needs to be done with the 2013 data, whit apparently strongly diverging results (copepods apparently do NOT contribute to the signal). Line 308: To which depth do the Euphausids migrate to? Hypoxic water? Figure 6: The material and methods say that the signal of ADCP and Simrad were combined (38 and 120 kHz). Which signals were used for the along fjord transects? Is the analysis comparable to the fixed stations? Line 318: ... demonstrated a uniform distribution of zooplankton'. This statement implies that the echo sounder provides a quantitative estimate of the groups studied, which is very likely not the case (see comments above). The authors should be more careful. He signal does not show any variation. Line 321: Why figure 6? Line 325: The NASC in the small figure included in Fig 6 is barely readable. How was the signal for fish and zooplankton obtained? The methods do not provide sufficient detail. There seem to be little differences at greater depth depending on the stations. Line

**OSD**
342: this is interpretation of the results, and should not be presented here. Again, I advise to avoid the general term zooplankton because the back-scattering likely represent only a part of the zooplankton. This needs to be extracted from the ADCP and zooplankton sampling. Line 347: The methods state that zooplankton was analysed in August 2014. The data is not presented. Again, avoid to assign zooplankton in general to the backscattering signal. Figure 7: the legend says zooplankton and fish, the figure shows 38FL, 38 BN and fish. How was noise identified? Line 359: Describing the signal is not a confirmation. Line 362: The in-situ (nets?) data is not shown. How can Euphausiids attributed to the signal? Line 371: What is meant by 'in-situ plankton sampling'? The echo-sound data? Zooplankton sampling with nets was conducted. The data is not shown. Line 373: The analysis needs explanation in the introduction and the methods. Why is a correlation between Sv and T to be expected? Fig 10 e and f: The methods do not describe how the measurements of energy dissipation with a resolution of 1mm where integrated to match the resolution of the backscatter analysis of 1m. Fig 11: Why is this presented?

Discussion: There is quite some literature on the relationship of zooplankton distribution in relation to oxygen minima and the relationships of zooplankton distribution and echo-sounder signals. These need to be explored. What do the present results add to these studies? The authors use their own data very little to explore the identity of the backscatter signal and to provide an analysis of general interest about the influence of physical factors and zooplankton beyond a local description. This needs to be conducted before any conclusions on implications of their relation to turbulence and implications for vertical flux can be made. Line 508: How the authors come to the conclusion that copepods cause the backscattering signal in the deep, hypoxic layer is unclear to me.

---

## Referee Comment (RC3) · Anonymous Referee #3 · 19 Dec 2017

Summary: The article analyses a dataset of ADCP, echosounder, CTD, turbulence and biological data in a fjord which is showing suboxic conditions in the deeper water column. The ADCP, echo sounding and biological data show a clear daily vertical migration pattern within the upper 100 m of the fjord. Turbulence measurements in the main fjord and in the Jascaf channel show different regimes, with strongly increased levels of turbulence in the Jascaf fjord. The authors try to correlate oceanographic conditions with abundance of zooplankton and its daily vertical migration.

Comment: While this is a very nice combined dataset of physical parameters and biology the processing and conclusions from this work have to be more elaborated before

its ready for publication. One fundamental parameter used is the relative abundance of zooplankton derived from accoustic backscattering. To my understanging the authours have the data to calculate this correlation by using the data in Fig. 4 and 5. It is not clear what turbulence data is used in this article. While there are two device (SCAMP and VMP-250) the data suggest that only the VMP-250 is used (Fig. 10 and 11), that has to be clarified. Temperature microstructure is problematic in low as well as high turbulence regions, I wonder which device was used where. I would also like to see example temperature microstructure profiles with examples of fitted data, showing that the fit is reasonable. I can imagine that the temperature microstructure has problems in the deeper part of the Puyuhuapi Fjord (Fig. 10c) as well as in the extremely high dissipation region in the Jacaf channel (Fig. 10d). Since tides are usually an important energy input for mixing, a section containing informations about tides is neccessary. Without a proper discussion I see no point in correlating all sorts of parameters against abundance of zooplankton (Fig. 9). The correlation does basically show that the zooplankton stays in the oxygenated water, which is already visible from the echo sounding transects. For a person who is not familiar with DVM, it is from the article itself not clear, why zooplankton should migrate at all, a discussion about the reasons is needed. Vertical oxygen concentrations are not steadily decreasing towards deeper layers, Fig. 2f shows, that towards the bottom oxygen increases again, are there reasons for that? A parameter which was not discussed at all is nitrate: There are nitrateclines, its hard to see if they are coinciding with the thermocline or halocline. Has nitrate a connection to zooplankton? Maby via phytoplanktion? There is data from different seasons, is there a seasonality? The abundance (O(4000 ind m-3)) of May Fig. 4 seems to be much higher than in January (Fig. 5, O(200 ind m-3)). In the introduction it was stated that the difference between the two echo sounding frequencies is used, the figures do anyhow show both frequencies separated (Fig. 7, 8), why is it so? Phytoplankton was not really discussed through the article but is mentioned in the conceptual figure and briefly in the discussion. Are the any hints about the abundance and temporal evolution of it? Fig. 12 also neglects that higher mixing might also deepen the mixed layer. A
comparison of vertical profiles of the VMP directly above the sill and in the fjord would be instructive.

Details: Fig. 2: A conceptual vertical profile at different position is needed. Two many profiles are on top of each other.

Fig. 3: Scale of salinity can be changed, the lower 15 gkg-1 are not used. Fig. 10 e+f: What does this correlation say?

---

## Referee Comment (RC4) · Anonymous Referee #4 · 28 Dec 2017

Review on "Turbulence and hypoxia contribute to dense zooplankton scattering layers in Patagonian Fjord System" (os-2017-89) by I. Perez-Santos, L. Castro, N. Mayorga, L. Ross, L. Cubillos, M. Gutierrez, E. Niklitschek, E. Escalona, N. Alegria and G. Daneri

The authors conducted an extensive field campaign to survey DVM of zooplankton in Patagonian Fjord combined with various physical parameters. The approach is correct, but the interpretation of the data, as well as the experimental design are not suited for the purpose. I have read three other referee's comments and I totally agreed with those comments. The context is poorly organized and too many references are missing from the reference list. Turbulence measurements are conducted with two different

instruments, but no data for SCAMP was presented in the text. The description of SCAMP should be deleted.

As I mentioned, I agree with the other reviewers' comments, I am not going to repeat the same points. But one of the major fraud should be repeated. 38 KHz is too low to detect zooplankton. In general, a combination of 38 KHz and 120 KHz is useful to distinguish between zooplankton and fish. Another important error that was not mentioned in the other reviews is that the dissipation rate estimate reached an upper bound at ∼5x10ˆ-5 W/Kg since, probably, they did not correct the unresolved variance in high wavenumbers (see Fig.10f). But they are reporting that the dissipation rates around sill are nearly 10ˆ-4 W/Kg. I see no reason to support this number. Also I do not see Kho=10ˆ-2 (mˆ2 sˆ-1) in Fig. 11c. All values are below 10ˆ-2!

Please also note the supplement to this comment:
https://www.ocean-sci-discuss.net/os-2017-89/os-2017-89-RC4-supplement.pdf

––––––––––––––––––––––––––––––––––

---

## Author Comment (AC1) · 15 Mar 2018

Summary Perez-Santos et al. use a dataset comprised of hydroacoustics coupled with hydrographic measurements and zooplankton samplings to examine the effects of physical and chemical properties on zooplankton distributions in a Patagonian Fjord. Interdisciplinary data collected in this study is remarkable, covering biological, physical, and chemical properties. The authors achieved extensive coverage both temporally

and spatially by combining moored and ship-based surveys. However, this manuscript lacks clear objectives and significance of the study. As a result, I cannot comment on the significance of the study. The Introduction contains specific information about the Patagonian fjords, which are more relevant to the section titled "study area", without placing the study to the larger context. In addition, the paper does not do an adequate job analyzing the echosounder data. While the paper focuses on the zooplankton distribution, frequency of echosounder chosen (i.e., 38 kHz) is not relevant to examine zooplankton. Although the observations may be of interest, this manuscript is not yet ready for publication.

General comments Abstract does not contain the objectives of this study, instead heavily focused on the methods. There is fairly detailed description about the study site, which is more appropriate to place in the main text.

We have changed the wording of the abstract to contain the study objectives and have removed most of the verbiage about the methods. We have moved the description of the study site from the introduction to the study area section.

Introduction should include the objectives and significance of the study. Currently, there is not enough description on knowledge gap in this field based on previous studies and justification of the study site. Detailed description of the study site is more appropriate to place in the Methods.

We have restructured the introduction to include a more comprehensive literature review, explanation of the research gap, presentation of the specific goals of this work, tools used to accomplish these goals and an outline of the coming sections. We have moved the detailed description of the study site to the study area section.

I have major concerns on the analysis of echosounder data to extract zooplankton backscatter. In general, zooplankton species found in their net samples (e.g., copepods, euphausiids) cannot be detected at 38 kHz, because they are too small to be significant backscatterers compared to the wavelength of 38 kHz. How did the authors

separate zooplankton from fish (Fig. 6)?

We added a new methodology section to the text (3.2.1 Echo-sounder data analysis) and some figures to address concerns of reviewer 1. Below is the section that was added to the manuscript:

3.2.1 Echo-sounder data analysis Post-processing of echo-sounder data was performed in Echoview (Myriax inc, Tasmania, https://www.echoview.com/), using the open access version ("FishZpkPeru38&120.evi") of Ballón et al., (2010)'s algorithm, which combines mean volume backscattering (MVBS) from 38 and 120 kHz, using both ∆MVBS (differences) and (summations) to discriminate and quantify the abundance of macrozooplankton. This algorithm separates raw data into three different virtual echograms: fish and two macrozooplankton groups (macrozooplankton or "fluid-like" and gelatinous or "blue noise" organisms). The fluid-like group follows a sphere model (Holliday & Pieper, 1995) considered appropriate to represent cylindrical and spherical shapes, including euphausiids and large copepods, which are dominant macrozooplankton groups off Peru and Chile (Ayon et al., 2008). The algorithm is considered to be useful for 38 and 120 kHz data from targets whose radius is ≥0.5 mm and has a dB difference of 2-19 dB (Ballón et al., 2010 and 2011). As implemented, the post-processing file FishZpkPeru38&120.evi is also designed to remove blind areas, near field, background noise and rainbow phenomenons. Given physical limitations imposed by sound absorption of selected frequencies (38 and 120 kHz) across the water column, an effective sampling of the water column up to 250 m was expected. Absorption is greater for 120 kHz, which exhibits the shortest range, but has a greater vertical resolution than 38 kHz. The 38 kHz frequency, on the other hand, exhibits a longer range, but limited resolution affecting small zooplankton (e.g. small copepods) detection. Nonetheless, this is the most commonly used frequency, which has proven to be efficient for studying macrozooplankton groups such as siphonophores, chaetognaths and euphausiids (Mair et al., 2005; Cade and Benoit-Bird, 2015; Ariza et al., 2016). Volume backscattering strength (Sv, dB re 1 m-1) values were integrated using a grid

of 20 m (depth) by 50 m (distance), and re-scaled into the customary index "nautical area scattering coefficient" (NASC, in units of m2 n mi2). Since NASC lies on the linear domain, it can be considered proportional to and suitable for indexing zooplankton abundance (Ballón et al., 2011). Ballón, M.: Acoustic study of macrozooplankton off Peru: biomass estimation, spatial patterns, impact of physical forcing and effect on forage fish distribution. These. Universite Montpellier II, 205 pp, 2010. Ballón, M., Bertrand A., Lebourges-Dhaussy A., Gutiérrez M., Ayón P., Grados, D., Gerlotto F.: Is there enough zooplankton to feed forage fish populations off Peru? An acoustic (positive) answer. Prog. Oceanogr., 91(4): 360-381, 2011.

When they have two frequencies (following Ballón et al. 2011), Sv data from 120 kHz is useful for up to _ 200 m depth due to the increase in background noise with range. However, the data analysis was conducted up to _ 450 m depth (Fig. 7). Only data within the analysis limit should be examined.

That is correct, the Ballón et al., 2010 algorithm shouldn't be used below 250 m (or even 200 m) if the purpose is quantitative. If used at depths below these the results become biased.

There is no discussion on the seasonal change in zooplankton distributions and compositions. Based on their seasonal coverage of the data sets, seasonal component should be considered in addition the difference in study sites.

We include a new discussion in the manuscript that incorporated the description of the seasonal behavior of zooplankton.

5.4 Other findings and considerations Results showed similar groups of macrozooplankton (>5 mm) in Puyuhuapi Fjord and Jacaf Channel: euphausiids, chaetognaths, medusae and siphonophores during summer (January 2014) and winter (winter 2014). However, euphausiids were not observed in fall 2013, which was an unexpected result which deserves further confirmation and analysis. In contrast, fall 2013 sampling presented the highest acoustic abundances within the time series (Fig. 3). The elevated

accumulation of zooplankton species around the sill may impose a significant modification in the amount and quality of carbon exported to deeper waters in particular zones of the fjords. Future studies on carbon flux quantification in fjords should incorporate sill regions to test this hypothesis, in order to improve ocean pumping assessments in the context of climate change and variability.

The manuscript needs to be carefully reviewed for typos and grammatical errors. Comments from co-authors (lines 710-712) remained in the main text, which need to be removed. Section numbers are not in sequence in the Results, Discussion, and Conclusions.

We eliminated grammatical errors throughout the text and organized the sequence of Results, Discussion, and Conclusions.

Technical comments

Description of the data collection is complicated and hard to follow because there are many sensors deployed during different times of the year at different locations. Inclusion of a table summarizing the details (e.g., types of data collected, deployment locations/depth, period of data collection) would increase the readability of the manuscript.

We included Table. 1 to better describe the the different oceanographic field campaigns. The new table is given below:

The method lacks detailed description of the echosounders, such as ping rate, calibration information, and preprocessing of the data (e.g., bottom detection, near-field removal, background noise removal).

The deployed echo-sounder is a Kongsberg Simrad EK60 operating 2 split beam type transducers; the data produced is RAW format and contains power Sv and TS values in addition to angle coordinates of peak values at the depth of every sample. The calibration was made by using proper cupper spheres and procedure contained in the handbook of the echosounder.

We added new information to the Data and methodology section in the manuscripts in order to better describe the echo-sounder measurements and methods (Section 3.2.1):

Use of the word, echosounders, should be consistent throughout the text. The authors use "echo sounders", "echo-sounders", and "echosounders", which need to be fixed.

We now use the word "echo-sounder" throughout the text.

Sv units, dB re 1 mȨ̈Ę-1, should be used throughout the text. The authors often use "dB" toward the end of the manuscript and figure legends and captions.

We changed Sv units in "dB" to "dB re 1 mȨ̈Ę-1" throughout the text.

Figures: Fig. 1: Color should be consistent between two colorbars. In the manuscript, red is SHALLOWER depth in the overview map, while red is DEEPER depth in zoomin figure, which are confusing. Some symbols overlap each other, which makes the readers difficult to understand the legends.

We eliminated the regional map from the figure to avoid confusion. We separated symbols. See new figure 1.

Fig. 2: Content of the figures on the top row overlaps with Fig. 3, and the patterns of the Jacaf Channel are very similar to those in the Puyuhuapi Fjord. Also, data points from previous studies are not discussed in the text. Thus, this figure could be removed from the manuscript.

We deleted figure 2 from the manuscript.

Fig. 3: Define "MSAAW", "SAAW", and "ESSW" in the figure caption. No definition of MSAAW and ESSW is stated in the text either. X-axis should be distance from the mouth, instead of latitude, because the fjord is positioned diagonally.

We added the complete name of water masses in subplot (c). We changed the x-axis distance from latitude to distance in km. See new figure 2.

Fig. 4: What does "AFIOBIOEX" mean? This term is not introduced in the text, but only appears in the figure captions (Fig. 5 as well). To improve the readability of the manuscript, AFIOBIOEX should be removed from the captions.

We eliminated AFIOBIOEX from the text. The new figure captions reads:

Figure 3. (a) Volume backscattering strength (Sv,) calculated from the ADCP-1 backscatter signal in Puyuhuapi Fjord, deployed at 50 m depth from the 8th to the 26th of May, 2013. (b) Zoom of the Sv data and the times of in-situ zooplankton sampling (black dots) carried out during May 25-26, 2013. (c) Vertical abundance of main zooplankton groups (>5 mm length) from the in-situ sampling at 18 h on May 25th and (d) 11 h on May 26th.

Fig. 5: The bars showing the standard deviation are not legible in (c)-(e). There is no x- and y-labels.

We eliminated this subplot from the figure and added new subplot (See new figure 4)

Fig. 6: X-axis of (a, c), and (d, f) is not consistent. All figures should be corrected for distance from the same reference point (e.g., distance from the mouth). What do the numbers in (b) and (e) mean? The upper bound of the hypoxia layer needs to be included, because it is not clear where the hypoxic layer is located.

We changed x-axis in subplots (d) and (f) to represent the same direction shown in (a) and (c). The numbers in (b) and (e) represent an index of zooplankton abundance (NASC: nautical area scattering coefficient) used in other manuscripts to estimate and quantify zooplankton biomass. We explain this now in the main body of the manuscript. Below are references for NASC:

Ballón, M., Bertrand A., Lebourges-Dhaussy A., Gutiérrez M., Ayón P., Grados, D., Gerlotto F. Is there enough zooplankton to feed forage fish populations off Peru? An acoustic (positive) answer. Prog. Oceanogr., 91(4): 360-381. 2011. Klevjer, T. A., Irigoien X., Røstad A., Fraile-Nuez E., Benítez-Barrios V. M. and Kaartvedt S. Large

scale patterns in vertical distribution and behaviour of mesopelagic scattering layers. Sci. Rep. 6, 19873; 2016. Sato, M., Horne J., Parker-Stetter S., Essington T., Keister J., Moriarty P., Li L., Newton J.: Impacts of moderate hypoxia on fish and zooplankton prey distributions in a coastal fjord. Mar. Ecol. Prog. Ser, Vol. 560: 57–72, 2016.

We included a new subplot in figure 6 (g) to better show the position of the hypoxic boundary layer and the hypoxic layer. The dissolved oxygen data was obtained between day and night-time acoustic sampling using continuous CTD profiles carried out approximately every 3 hours during January 23-24, 2014. We also added in subplots (a) and (d) the position of the hypoxic boundary layer. See new figure 5

Fig. 7: There is no need to plot the same data at two different frequencies. 38 kHz for fish and 120 kHz for zooplankton are commonly used in bioacoustic field.

We changed Fig 7 and Fig 8. See new figure 6 and 7.

Fig. 9: Which frequency is used for Sv values?

We used 38 kHz. We now make this clear throughout the main body of the manuscript.

Below is some examples of typos: Remove a period from the title. Line 69: change "has" to "have". We changed 'has' to 'have' on line 69.

Line 98: add comma after "advection". We eliminated this sentence from the Introduction

Line 104: add comma after "CTD profiles".} We added comma after "CTD profiles"

Line 128-129: "northern mouth" cannot be identified in Fig. 1, because the subset of the figure blocks the portion of the fjord map.

The new Figure 1 shows the northern mouth.

Lines 143: Be consistent for the use of numbers (e.g., one vs. 1).

We changed the sentence.

Line 205: What does "CITA" mean? We eliminated CITA from the text.

Line 210: Ballón et al. (2011) is not in the References. We added the reference of Ballón et al., (2011) to the reference list.

Ballón, M., Bertrand A., Lebourges-Dhaussy A., Gutiérrez M., Ayón P., Grados, D., Gerlotto F.: Is there enough zooplankton to feed forage fish populations off Peru? An acoustic (positive) answer. Prog. Oceanogr., 91(4): 360-381. 2011.

Line 210: Unit of NASC is "mËĘ2 nmi-2". The equation of NASC does not need to be presented, because this is a common knowledge. We eliminated the NASC equation from the text.

Lines 240-241: Remove the references, because these are commonly used techniques. We removed the references and also removed the reference of Castro et al., 2011 from the reference list.

Lines 243-247, 277-279: There is no need to include the study plan that did not happen. We eliminated this sentence from the text.

Line 266: What does "ESSW" mean? We clarify the mean of ESSW in the text as: Equatorial Subsurface Water (ESSW)

Line 274: Change "<" to ">". We changed symbol in the text.

Lines 336-337: There is no time on the x-axis of Fig. 6. Time should be included on the x-axis, so that the readers can follow your interpretation. We included the information of the x-axis in the text.

Line 371: Change "+" to "and". "DO" should be defined and used throughout the text, instead of using both DO and dissolved oxygen.

We defined Dissolved Oxygen (DO) in the Introduction section and changed dissolved oxygen to DO throughout the text.

Line 414: "Others" should be "other".

We changed 'Others' to 'other'.

Lines 425-428: This sentence is contradicting. Did you mean there is twilight vertical migration, or not? We clarify this in the new discussion section.

Line 469: Remove "(Fig. 10)". This is duplication. We removed Fig. 10 from the text

Line 480: Cut "in" before "there might be". We eliminated "in" from the text.

Please also note the supplement to this comment:
https://www.ocean-sci-discuss.net/os-2017-89/os-2017-89-AC1-supplement.pdf

[Figure]

[Figure]

**Fig. 1.** Study area in relation to South America and the Pacific Ocean is the small panel in the top right. The main figure enlarges the study area (Puyuhuapi Fjord and Jacaf Channel) and indicates the instru

[Figure]

[Figure]

**ADCP-1 (307.2 kHz)**  (a)

Depth (m)

Time (days, May 2013)

Sv
(dB re 1 m$^{-1}$)

May 25-26, 2013  (b)

Depth (m)

18:00  (c)

11:00  (d)

Depth(m)

**Zooplankton >5mm**
- Siphonophores
- Medusae
- Chaetognaths

Abundance (ind m$^{-3}$)

**Fig. 3.** (a) Volume backscattering strength (Sv,) calculated from the ADCP-1 backscatter signal in Puyuhuapi Fjord, deployed at 50 m depth from the 8th to the 26th of May, 2013. (b) Zoom of the Sv data and the

[Figure]

**Fig. 4.** (a) Volume backscattering strength (Sv) calculated from the ADCP-2 backscatter signal in Puyuhuapi Fjord from the 22nd to the 24th of January, 2014. The in-situ zooplankton sampling (in 3 h intervals)

**38 kHz**

**Fig. 5.** (a) Volume backscattering strength (Sv) calculated from the ADCP-2 backscatter signal in Puyuhuapi Fjord from the 22nd to the 24th of January, 2014. The in-situ zooplankton sampling (in 3 h intervals)

[Figure]

**Fig. 6.** (a) Scientific echo-sounder transects along Puyuhuapi Fjord (0-18 km) and Jacaf Channel (18-35 km)on August 17, 2014 using the combination of 38 and 120 kHz frequency. (a) Fluid like and (b) blue nois

**120-38 kHz**

**August 18, 2014**

Sill

Fluid like **(a)**

Blue noise **(b)**

Distance (km)

Sv
(dB re 1 m⁻¹)

-75   -80   -85   -90   -95   -100

Fish **(c)**

Distance (km)

Sv
(dB re 1 m⁻¹)

-35   -40   -45   -50   -55   -60   -65   -70   -75

**Fig. 7.** Acoustic transect over Jacaf sill using the combination of 38 and 120 kHz frequency on August 18, 2014. (a)Fluid-like echogram, (b) blue noise echogram for zooplankton and (c) the fish echogram. Distr

[Figure]

[Figure]

[Figure]

**Fig. 8.** (a) Depth integrated abundance of zooplankton groups from surface to 150 m depth for various sampling hours (b) euphausiids contined in depth strata (mean and standard deviation) during daytime (red)

[Figure]

Fig. 9. Relationships between the relative abundance of zooplankton (expressed in Sv values) using 38 kHz frequency echo-sounder measurements (y-axis) and temperature in (a) Puy-huhuapi Fjord and (b) Jacaf Ch

[Figure]

**Fig. 10.** Profiles of water temperature (blue line), vertical shear (red line) and dissipation rate of turbulent kinetic energy (black line with green dots) obtained with the VMP-250 microprofiler at the depth

**Fig. 11.** (a) Microstructure profile locations along Jacaf Channel and sill using VMP-250 in November 2013. (b) The color bar showed the dissipation rate of turbulent kinetic energy ($\varepsilon$) and the blue lines depi

[Figure]

**Fig. 12.** Conceptual model to show the oceanographic processes that contribute to the distribution and aggregation of zooplankton in (a) Puyuhuapi Fjord and (b) Jacaf Channel.

Table 1. Data set collected during oceanographic campaigns in Puyuhuapi Fjord and Jacaf Channel.

| Location | Date | Season | Data measured | Instruments |
|---|---|---|---|---|
| Puyuhuapi Fjord | May 8-27, 2013 | Fall | -Acoustic data 300 kHz | -ADCP-1 RDI |
| | | | -Zooplankton | -WP2 net |
| | January 22-25, 2014 | Summer | -Acoustic data 300 kHz | -ADCP-2 RDI |
| | | | -Acoustic data 38 kHz | -SIMRAD EK60 |
| | | | -Zooplankton | -Tucker Trawl net |
| | | | -Turbulence | -VMP-250 |
| | | | -Hydrography | -YSI 6600 |
| | August 17-19, 2014 | Winter | -Acoustic data 38 and 120 kHz | -SIMRAD EK60 |
| | | | -Zooplankton | -Tucker Trawl net |
| | February-June 2016 | Summer-Fall | -Tidal data (south) | -HOBO U20 |
| | February-November 2016 | Summer-Spring | -Tidal data (north) | -HOBO U20 |
| | June 16, 2016 | Fall | -Hydrography | -CTD SBE-25 |
| Jacaf Channel | April-November 2012 | Fall-Spring | -Tidal data | -HOBO U20 |
| | November 21, 2013 | Spring | -Turbulence | -VMP-250 |
| | August 2014-May 2015 | Winter-Spring – Summer-Fall | -Tidal data | -ADCP-3 |
| | August 17-19, 2014 | Winter | -Acoustic data 38 and 120 kHz | -SIMRAD EK-60 |
| | | | -Zooplankton | -Tucker Trawl net |

**Fig. 13.** Table 1. Data set collected during oceanographic campaigns in Puyuhuapi Fjord and Jacaf Channel.

Table 2. Harmonic analysis implemented to water  level time series in Puyuhuapi Fjord and Jacaf Channel.

| Sea level time series | Date (mm-yyyy) | Energy from semi-diurnal band (m² cph⁻¹) | Amplitude of principal constituents (cm) | | | | F | Tidal regime |
|---|---|---|---|---|---|---|---|---|
| | | | $M_2$ | $S_2$ | $O_1$ | $K_1$ | | |
| Jacaf-HOBO | 04-09/2012 | 45.10 | 83.45 | 28.32 | 14.46 | 22.33 | 0.32 | Mixed semi-diurnal |
| Jacaf-ADCP | 08/2014-05/2015 | 57.29 | 60.67 | 61.01 | 57.78 | 42.48 | 0.82 | Mixed semi-diurnal |
| Puyuhuapi-HOBO south | 02-06/2016 | 44.45 | 81.97 | 31.51 | 13.37 | 18.36 | 0.27 | Mixed semi-diurnal |
| Puyuhuapi-HOBO north | 02-11/2016 | 49.17 | 89.15 | 31.07 | 11.03 | 17.75 | 0.23 | Semi-diurnal |

**Fig. 14.** Table 2. Harmonic analysis implemented to water level time series in Puyuhuapi Fjord and Jacaf Channel.

---

## Author Comment (AC2) · 15 Mar 2018

The manuscript aims at relating the vertical distribution and migration of zooplankton to physical structures and turbulence in Chilean fjord system. This is a timely and interesting focus. However, I find the manuscript in its present form preliminary and of local interest only. The difficulties are: The objectives of the study appear primarily of technical nature. The relevance and implications of studying the vertical distribution in

relation to fine scale properties and turbulent mixing needs to be highlighted in more detail. The introduction and the discussion basically lack scientific questions related to the physical-biological interactions and do not relate to an already large body of literature about detecting zooplankton with acoustic methods or the influence of physical (turbulence) or chemical (oxygen-minimum zones) properties on zooplankton distribution. The implied effects on reproduction, growth and life cycles in the introduction are not sufficient and appear redundant because the physical and biological processes occur on very different time cycles. Reference to previous work is largely restricted Chilean fjords. In addition, the material and methods are incomplete and inconsistent. Many details can be found below. It is unclear to me, why hydrographical data from 1995-2015 is presented, while zooplankton sampling is restricted to a few occasions. Data on zooplankton from net sampling in August 2014 is not presented although samples were apparently taken; instead physical data from 2016 is presented although not described in the Methods. Finally, the authors make very little use of their own data, particularly with regard to the identification of the primary groups responsible for the detected backscattering signals. The zooplankton depth resolved data should be presented and analysed. Data from 2013 suggests that copepods contribute very little to the signal, but the authors treat the backscatter data as equivalent to zooplankton throughout the manuscript. Tremendous differences in the abundance of zooplankton despite similar backscatter signal strength needs to be explained. In its present form, I cannot recommend considering the manuscript for publication and suggest that the authors revise it considerably.

Detailed comments:

Introduction • Line 70: Palma (2008) is missing the reference list We added the reference of Palma (2008) to the reference list:

Palma S.: Zooplankton distribution and abundance in the austral Chilean channels and fjords. Progress in the oceanographic knowledge of Chilean inner waters, from Puerto Montt to Cape Horn. Comité Oceanográfico Nacional - Pontificia Universidad Católica

de Valparaíso, Valparaíso, Chile, pp. 107-113. Book on line at http://www.cona.cl/, 2008. • Line 74: Landaeta et al. (2013) is missing in the reference list. When microzooplankton and fish larvae were studied, copepods (meso-and macrozooplankton) cannot dominate.

We added the reference of Landaeta et al., (2013) to the reference list:

Landaeta M., Martínez R., Bustos C. and Castro L.: Distribution of microplankton and fish larvae related to sharp clines in a Patagonian fjord. Revista de Biología Marina y Oceanografía, Vol. 48, N°2: 401-407, 2013.

As we mentioned in the text, microzooplankton and fish larvae were studied in Steffen fjord (-47.4° S). See new Introduction section.

• Line 80 following: Rephrase the sentence. Why 'although'? What is meant by accurate results? Nets and acoustic methods provide principally different results with high taxonomic resolution in the first and high spatial resolution in the second. Thus, they are used to study different aspects and differ largely in their size resolution. We eliminated this sentence from text.

• Line 88: Please specify: Norwegian Channel or Kattegat? We clarify sentence in the new Introduction section

• Line 89: Buchholz et al. 1995, Zhou and Dorland 2004 are missing in the reference list.

We added the references to the reference list:

Buchholz F., Buchholz C., Reppin J., Fischer J. Diel vertical migrations of Meganyctiphanes norvegica in the Kattegat: Comparison of net catches and measurements with Acoustic Doppler Current Profilers. Helgolander Meeresunters, 49, 849-866, 1995. Zhou M., Dorland R. Aggregation and verticalmigration behavior of Euphausia superba. Deep-Sea Res. II 51, 2119–2137, 2004.

• Line 89 following: The necessity and need for studying the vertical distribution or migration in relation to physical properties needs to be described better. They are themselves not a scientific question.

We explicitly state this now in the modified Introduction section.

• Line 97: Please specify the implications for reproduction and growth. Yamasaki et al. 2002 is missing in reference list. We eliminated this paragraph from the text.

• Line 101: The influence of the described processes (short-term) on biological life cycles (different time scales) needs to be explained.

The Introduction section was re-organized and this sentence was deleted.

• Line 108: What is meant by 'survival strategies present in these organisms'?

We eliminated this sentence from the text. • Line 111: It is unclear to me what the stage-specific migration patterns of Rhincalanus have to do with the effect of fine scale turbulence patterns. Acoustics cannot be used to resolve the stages of this species. • We eliminated this paragraph from the text.

• Line 115: The introduction lacks a review of the present knowledge about the effect on turbulent mixing and the oxygen conditions on the distribution of zooplankton. What are the scientific questions? What zooplankton is targeted at? Nets and acoustic profilers provide largely different type of data. • We have addressed this in the modified Introduction section.

Material Methods • Line 146: Specify the depths for nutrient samples. We eliminated the information on nutrients from the manuscripts.

• Line 201: The description of the echo sounder needs to be checked. On line 188, it says SIMRAD CX 34 at 38 kHz, here it says EK-60 at 38 and 120 kHz. Please specify also how the echo intensity was combined. We added table 1 to clarify the sampling program and instruments used in each field campaign.
Table 1. Data set collected during different oceanographic campaigns in Puyuhuapi fjord and Jacaf channel. We added a new section to clarify the Acoustic methodology. See 3.2.1. Acoustic data analysis from echo-sounders

• Line 210: Please explain the units (what is n and mi<square>. Zooplankton abundance is usually presented per unit volume, thus the full units should be presented (also of T)

The units of m2 n mi2 using to the nautical area scattering coefficient (NASC) is an acoustic unit used as an index of zooplankton abundance and it's not comparable to zooplankton abundance (Ind m-3) obtained with sampling nets. The unit of T is meter.

The NASC formula was eliminated from the text as was recommended by RC#1.

• Line 217: What is 'Tx' in the formula?

Tx is the temperature at the transducer (°C) and is now included in the text.

• Line 238: The sampling in 2013 covered only the upper 50 m but not the water column of 100 m scanned by the ADCP. Why? During May 2013 ADCP-1 was moored at 50 m depth to study the near-surface velocities of the fjord. This mooring was not orientated to the DVM of zooplankton research, but the backscatter data showed the first record of DVM in Puyuhuapi Fjord and then motivated the study of zooplankton using acoustic techniques.

• Line 249: No information is presented on the analysis of the sampling. From the data presented in the study no differentiation into size classes was performed. Why?

We clarified in a new section (3.3 Zooplankton sampling).

Results: • Line 254 following: In Fig 2, the top 100 m should be resolved because the size of the graphs make it very difficult to extract the information on T, S, and the other variables. The legend should be self-explanatory, but it is not. The T at the surface is 15 degrees, the x-axis stops at 14 degrees. The text and figures do not

always match: during Puy V hypoxic water occurred at a depth > 200m and not as implied by the text at >100 m. We eliminated figure 2 as was recommended by RC#1.

• Line 263: The Mat Meth indicate that the sampling covered the period 1995-2015. Now data from 2016 are presented. This is confusing. Why was this data included? We clarified the information in the new section 3.1 Water column properties

• Line 275: Sampling was conducted in layers of 10 m depth, but data on zooplankton distribution is averaged. Why? Information on size classes should be presented. In addition: was the abundance integrated as indicated in the figure legend? Then m-3 is wrong.

The data were presented as integrated to show the variation in abundance throughout time, and in particular, the increase in abundance during the first night hours, that we believe correspond to the start of the vertical migration upwards (e.g. at. 20.00h). The increase in zooplankton abundance resulted from their ingress from deeper layers during daytime hours. Integrated abundances may be expressed either as ind x m-2 or ind x m-3 (when divided by the depth of the water column sampled, 50 m at all sampling times in this case). We now explain this in the methods section In this new version of the manuscript we included the size classes of the zooplankton as requested by the reviewer and the vertical distribution (day vs. night) of siphonophores, chaetognaths and medusae (for example, as in the new figure 4). Euphausiids were not included in the new figure 3, because they were absent in most (all but one) samples probably because they were deeper and started to migrate upwards later at night than our last sampling hour at dusk.

-Fig. 4 c does not allow extracting quantitative information on siphonophores.

The data on siphonophores is shown now in the vertical distribution of the new figure 3 c-d (see previous answer)

-Figure 5: Apparently, zooplankton was analyzed in size categories; this needs to be

described in the methods.

This is now included in methods section.

A lot of information is lost by averaging/integration (this is not clear to me; it looks like averaging but integration is stated). I suggest to present the zooplankton data (size, taxa) as in Figure 5a despite a courser resolution. -Then, signal and zooplankton distribution can be compared.

The size data is now included and new subplots were added to show some examples of the vertical distribution. See new figure 4c-d.

Negative abundances in Fig c-e are odd. We eliminated this subplot from the figure.

-Zooplankton abundance in Jan 2014 (daytime) is several orders of magnitude lower than in 2013 (daytime), but signal strength appears similar or even higher. This needs explanation.

The zooplankton abundance of the larger size groups is lower in January 2014 than in May 2013 (3x or 4x) but not orders of magnitude. The largest differences are in copepods.

• Line 303: The authors describe here that copepods and others together contribute to the signal in backscatter. This is not conclusive until the data is shown in high resolution as described above. In addition, a similar analysis needs to be done with the 2013 data, whit apparently strongly diverging results (copepods apparently do NOT contribute to the signal). We clarify the sentence in the new manuscript.

• Line 308: To which depth do the Euphausids migrate to? Hypoxic water? The in-situ zooplankton sampling did not extend to the hypoxic water, which is why the results show the Euphausids migrating only in the first 100 meters of the water column.

• Figure 6: The material and methods say that the signal of ADCP and Simrad were combined (38 and 120 kHz). Which signals were used for the along fjord transects?

We used 38 kHz in the along fjords transects. We clarified this information inside the figure and in the figure caption. We also reiterate this in the manuscript text.

• Is the analysis comparable to the fixed stations? Line 318: ': : : demonstrated a uniform distribution of zooplankton'. This statement implies that the echo sounder provides a quantitative estimate of the groups studied, which is very likely not the case (see comments above). The authors should be more careful. He signal does not show any variation. We eliminated this sentence from the text.

• Line 321: Why figure 6? We have clarified this.

• Line 325: The NASC in the small figure included in Fig 6 is barely readable. How was the signal for fish and zooplankton obtained? The methods do not provide sufficient detail. There seem to be little differences at greater depth depending on the stations. We added new subplots to the new figure 5 (see above) that show the average values of NASC from zooplankton during daytime and night hours (Fig.5b and Fig. 5e). Also the average NASC of fish was also calculated and maximum NASC values were observed similar to the echogram, but NASC values were higher than NASC from zooplankton due to the difference in Sv magnitude. In this work the fish representation was only utilized to understand the prey-predator relationship, as we now mention in the Discussion section.

• Line 342: this is interpretation of the results, and should not be presented here. Again, I advise to avoid the general term zooplankton because the back-scattering likely represent only a part of the zooplankton. This needs to be extracted from the ADCP and zooplankton sampling. We clarified the sentence in the new text. • Line 347: The methods state that zooplankton was analysed in August 2014. The data is not presented. Again, avoid to assign zooplankton in general to the backscattering signal.

We clarified the sentence in the new text. In this new version of the manuscript we show the zooplankton data from August 2014 in Jacaf Channel (Figure 8). The zooplankton

data show increases in abundance during night hours (compared with daytime hours), the rising of zooplankton groups at night and most groups showed highest abundances at 100-150m during the daytime, which is deeper than in Puyuhuapi Fjord.

• Figure 7: the legend says zooplankton and fish, the figure shows 38FL, 38 BN and fish. How was noise identified? We added a new section to clarify the Acoustic methodology. "3.2.1. Acoustic data analysis from echo-sounders".

• Line 359: Describing the signal is not a confirmation.

We clarified the sentence in the new text.

• Line 362: The in-situ (nets?) data is not shown. How can Euphausiids attributed to the signal?

The new figure 8 showes this information.

• Line 371: What is meant by 'in-situ plankton sampling'? The echo-sound data? Zooplankton sampling with nets was conducted. The data is not shown. We clarified the sentence in the new text.

• Line 373: The analysis needs explanation in the introduction and the methods. Why is a correlation between Sv and T to be expected?

We added a description of the statistical methods applied to compare the signal Sv to environmental data, such as zooplankton groups, dissipation rate of turbulent kinetic energy ($\varepsilon$) and also $\varepsilon$ vs. zooplankton groups. The new information is now included in Section 3 Data and methodology section.

• Fig 10 e and f: The methods do not describe how the measurements of energy dissipation with a resolution of 1mm where integrated to match the resolution of the backscatter analysis of 1m. We eliminated this figure from text as recommended by R1.

• Fig 11: Why is this presented? Figure 11 was presented to evidence the intense

shear layers measured over the sill in Jacaf Channel. The direct measurements of shear allowed are linked to the high dissipation rate of turbulent kinetic found near the sill favoring vertical mixing and aggregation of plankton around the sill.

Discussion:

• There is quite some literature on the relationship of zooplankton distribution in relation to oxygen minima and the relationships of zooplankton distribution and echo-sounder signals. These need to be explored.

Thank you for the comment, it helps to clarify and highlight some of the results obtained in the study. New literature is now mentioned in the manuscript.

• What do the present results add to these studies?

As we now mention in the discussion section:

This study represents one of the first attempts to combine measurements of acoustics, stratified plankton sampling, microstructure profiles, and standard hydrographic profiles to investigate both the vertical distribution patterns of zooplankton and why these patterns exist in northwest Patagonian Fjords and other subantarctic latitudes. Three main findings resulted from this effort. First, DVM patterns of zooplankton became evident from all methodological approaches, at all study periods: May 2013, January 2014 and August 2014 (Fig. 3-8). Second, strong evidence arose showing zooplankton avoidance of hypoxic layers. And, third, a clear increment of zooplankton and fish aggregations around the Jacaf sill could be related to increased turbulence in this area.

• The authors use their own data very little to explore the identity of the backscatter signal and to provide an analysis of general interest about the influence of physical factors and zooplankton beyond a local description. This needs to be conducted before any conclusions on implications of their relation to turbulence and implications for vertical flux can be made.

We have now included day and night profiles of zooplankton vertical distributions during

all field campaigns. In addition, we have included data on the major zooplankton groups present in the fjord (by species and size) and provide more information on the type of backscatter signal used to differentiate between major zooplankton groups and fishes. Regarding aspects of general interest, in the previous answer we mentions how the manuscript now better describes our results.

• Line 508: How the authors come to the conclusion that copepods cause the backscattering signal in the deep, hypoxic layer is unclear to me.

We deleted this sentence

---

## Author Comment (AC3) · 15 Mar 2018

Anonymous Referee #3 Received and published: 19 December 2017

Summary: The article analyses a dataset of ADCP, echosounder, CTD, turbulence and biological data in a fjord which is showing suboxic conditions in the deeper water column. The ADCP, echo sounding and biological data show a clear daily vertical migration pattern within the upper 100 m of the fjord. Turbulence measurements in the

main fjord and in the Jascaf channel show different regimes, with strongly increased levels of turbulence in the Jascaf fjord. The authors try to correlate oceanographic conditions with abundance of zooplankton and its daily vertical migration.

**Comment:**

åĂć While this is a very nice combined dataset of physical parameters and biology the processing and conclusions from this work have to be more elaborated before its ready for publication. One fundamental parameter used is the relative abundance of zooplankton derived from acoustic backscattering. To my understanding the authors have the data to calculate this correlation by using the data in Fig. 4 and 5. It is not clear what turbulence data is used in this article.

The turbulence data were used in this article to justify the abundance of zooplankton around Jacaf sill. We believe that turbulence generated by tidal flow interacting with the shallow sill produced intense tidal currents and is the principal mechanism contributing to mixing in the fjord. As a result, this enhanced the nutrient availability to the phytoplankton, generating excellent conditions for the zooplankton and thus leading to increased aggregation in this area. This situation was not observed in Puyuhuapi fjord, where turbulence was less intense.

ć While there are two device (SCAMP and VMP-250) the data suggest that only the VMP-250 is used (Fig. 10 and 11), that has to be clarified.

We removed the SCAMP information's and data from the text.

We included Table. 1 to better describe the characteristics of the different oceanographic field campaigns (see Table 1 below).

Table 1. Data set collected during different oceanographic campaigns in Puyuhuapi fjord and Jacaf channel.

ć Temperature microstructure is problematic in low as well as high turbulence regions, I wonder which device was used where. We have now included the Table. 1 to better describe the characteristics of the different oceanographic campaigns and to detail which instruments were used during different campaigns. As we mentioned before the SCAMP data was removed from the text.

åÅć I would also like to see example temperature microstructure profiles with examples of fitted data, showing that the fit is reasonable. I can imagine that the temperature microstructure has problems in the deeper part of the Puyuhuapi Fjord (Fig. 10c) as well as in the extremely high dissipation region in the Jacaf channel (Fig. 10d).

We eliminated the old figure Fig. 10. As was mentioned by R3, SCAMP microstructure does not work well under strong tidal current conditions. Taking this into account, we decided to only include data from the VMP-250 (turbulence measured from velocity shears) from Puyuhuapi Fjord and Jacaf channel. See new figure 10.

ć Since tides are usually an important energy input for mixing, a section containing informations about tides is neccessary.

A new section was added that describes the tidal regime in Puyuhuapi Fjord and Jacaf Channel. See the new sections 3.4 and 4.5 below:

3.4. Tidal harmonic analysis The tidal constituents were computed using HOBO U20 water level loggers and the pressure sensor from ADCP-3 (Table 1-2, Fig. 1). A tidal harmonic analysis was applied to the sea level time series according to Pawlowicz et al., (2002), which considers the algorithms of Godin (1972, 1988) and Foreman (1977, 1978). We classified tides by the dominant period of the observed tide based on the form factor (F), defined by the ratio between the sum of the amplitudes of the two main diurnal constituents (larger lunar declinational, O1 and luni-solar declinational, K1) and the sum of the amplitudes of the two main semi-diurnal constituents (principal lunar, M2 and principal solar, S2), F = (O1+K1)/(M2+S2) (Bearman , 1989; where, F < 0.25 semi-diurnal, 0.25 < F < 1.5 Mixed semi-diurnal and F > 3.0 diurnal).

Table 2. Harmonic analysis implemented to water level time series in Puyuhuapi Fjord

and Jacaf Channel.

4.5 Tidal regime The harmonic analysis carried out with the sea level time series obtained in Puyuhuapi Fjord and Jacaf Channel, denoted the dominance (in terms of amplitude) of the semi-diurnal constituents (M2 and S2; Table 2). Diurnal constituents (O1 and K1) were also important, specifically at the Jacaf ADCP-3 station located close to the Jacaf sill region (Table 2 and Fig 1). The contribution of diurnal constituents added the mixed character to the tidal regimen in the study area. The spectral analysis implemented at all sea level stations showed maximum energy in the semi-diurnal band (Table 2), with the highest spectral energy (57.29 m2 cph-1) at Jacaf sill (Jacaf ADCP-3 station), which could be due to the extreme convergence of the channel at this location accelerating the tidal flows. aĂć Without a proper discussion I see no point in correlating all sorts of parameters against abundance of zooplankton (Fig. 9). The correlation does basically show that the zooplankton stays in the oxygenated water. which is already visible from the echo sounding transects. As this study is the first in Patagonian Fjords to establish a relationship between backscattering signals (Sv, proxy of zooplankton) with oceanographic variables, we believe it is important to show the temperature and salinity range where most of Sv values were observed. In the case of salinity, most of the Sv signal was located in oceanic water and not in estuarine water.

By correlating the different parameters we provide another way to show: that the zooplankton stay in oxygenated water.

aĂć For a person who is not familiar with DVM, it is from the article itself not clear, why zooplankton should migrate at all, a discussion about the reasons is needed. We include new information in the Discussion section to clarify the importance of DVM of zooplankton from Patagonian fjords and channels.

ć Vertical oxygen concentrations are not steadily decreasing towards deeper layers, Fig. 2f shows that towards the bottom oxygen increases again, are there reasons for that? We eliminated figure 2 from the text as was recommend by R2. The increase of DO values close to the bottom is due to deep ventilation processes that occur in this fjord. Pérez-Santos (2017), reported a deep ventilation event in the same area that helped to clarify and understand DO profiles in Puyuhuapi Fjord. The reference is: Pérez-Santos, I. Deep ventilation event during fall and winter of 2015 in Puyuhuapi fjord (44.6°S). Latin American Journal of Aquatic Research. Vol. 45(1). DOI: 10.3856/vol45-issue1-fulltext-25. ć A parameter which was not discussed at all is nitrate: There are nitrateclines, its hard to see if they are coinciding with the thermocline or halocline. Has nitrate a connection to zooplankton? Maby via phytoplanktion? We eliminated figure 2 from the text as was recommend by R1.

aÅć There is data from different seasons, is there a seasonality? The abundance (O(4000 ind m-3)) of May Fig. 4 seems to be much higher than in January (Fig. 5, O(200 ind m-3)). We included this information in the new Discussion section

ć In the introduction it was stated that the difference between the two echo sounding frequencies is used, the figures do anyhow show both frequencies separated (Fig. 7, 8), why is it so? We clarified this information in our response to R1 comments. These figures were changed. ć Phytoplankton was not really discussed through the article but is mentioned in the conceptual figure and briefly in the discussion. Are the any hints about the abundance and temporal evolution of it?

The phytoplankton studies in this region revealed seasonal behavior, represented by a productive season from August to April and a less productive season from May to July. The references are:

Daneri, G., Montero P., Lizárraga L., Torres R., Iriarte J.L., Jacob B., González H.E. and Tapia F.J.: Primary productivity and heterotrophic activity in an enclosed marine area of central Patagonia (Puyuhuapi channel; 44S, 73W). Biogeosciences Discuss 9, 5929–5968, 2012. Montero, P., Pérez-Santos I., Daneri G., Gutiérrez M., Igor G., Seguel R., Crawford D., Duncan P.: A winter dinoflagellate bloom drives high rates of

primary production in a Patagonian fjord ecosystem, Estuar. Coast. Shelf Sci., 199, 105-116, 2017a. Montero P, Daneri G., Tapia F., Iriarte JL. and Crawford D: Diatom blooms and primary production in a channel ecosystem of central Patagonia. Lat. Am. J. Aquat. Res., 45,(5), 999-1016, 2017b.

ć Fig. 12 also neglects that higher mixing might also deepen the mixed layer.

We changed figure 12. The new figure shows the position of the pycnocline deeper in Jacaf Channel than in Puyuhuapi Fjord. Also the nitrate and phosphate reference was eliminated. See new figure 12.

ć A comparison of vertical profiles of the VMP directly above the sill and in the fjord would be instructive.

A new figure was added to the manuscript to compare turbulence in Puyuhuapi Fjord and Jacaf Channel using the VMP-250 microstructure profiler.

ć Details: Fig. 2: A conceptual vertical profile at different position is needed. Two many profiles are on top of each other.

We eliminated figure 2 from the text as was recommend by R1. ć Fig. 3: Scale of salinity can be changed, the lower 15 gkg-1 are not used.

We changed figure 3 to the new figure 2.

ć Fig. 10 e+f: What does this correlation say?

We eliminated figure 10 from the text.

---

## Author Comment (AC4) · 15 Mar 2018

Anonymous Referee #4 Review on "Turbulence and hypoxia contribute to dense zooplankton scattering layers in Patagonian Fjord System" (os-2017-89) by I. Perez-Santos, L. Castro, N. Mayorga, L. Ross, L. Cubillos, M. Gutierrez, E. Niklitschek, E. Escalona, N. Alegria and G. Daneri The authors conducted an extensive field campaign to survey DVM of zooplankton in Patagonian Fjord combined with various physical parameters. The approach is correct, but the interpretation of

the data, as well as the experimental design is not suited for the purpose. I have read three other referee's comments and I totally agreed with those comments. The context is poorly organized and too many references are missing from the reference list.

We have taken into account all the referees comments (R1, R2 and R3) in order to enhance the manuscript quality. We believe that as a result the manuscript has improved tremendously, and we are grateful for the time of all of the reviewers in helping to better our manuscript.

Turbulence measurements are conducted with two different instruments, but no data for SCAMP was presented in the text.

Figure 10 was comprised of data from two microstructure profilers. In the left panel (Figure 10a, c, and e) we used the VMP data and in the right panel (Figure 10b, d and f) the SCAMP data was presented. In the revised version of the manuscript, after taking into account all of the reviewers comments, Figure 10 was eliminated from the text and an improved figure 10 was included that only uses the VMP-250 data.

The description of SCAMP should be deleted. As I mentioned, I agree with the other reviewers' comments, I am not going to repeat the same points.

We eliminated all information and data from the SCAMP.

But one of the major fraud should be repeated.

No fraud occurred in the manuscript.

38 KHz is too low to detect zooplankton. In general, a combination of 38 KHz and 120 KHz is useful to distinguish between zooplankton and fish.

We clarified the methodology as a result of R1 comments and new text was inserted in the revised version of the manuscript.

Another important error that was not mentioned in the other reviews is that the dissipation rate estimate reached an upper bound at âĹij5x10ËĘ-5 W/Kg since, probably, they

did not correct the unresolved variance in high wavenumbers (see Fig.10f). But they are reporting that the dissipation rates around sill are nearly 10Ȩ̈E-4 W/Kg. I see no reason to support this number. Also I do not see Kho=10Ȩ̈E-2 (mȨ̈E2 sȨ̈E-1) in Fig. 11c. All values are below 10Ȩ̈E-2!

Following the method of Lueck (2013) all of the shear estimates are cleaned, and noise is eliminated. The variance in high wavenumbers can be resolved. We eliminated old values of dissipation rate of turbulent kinetic energy and Kho throughout the text. The new values were added to the manuscript.
* * *

---

## Author Comment (AC5) · 15 Mar 2018

The comment was uploaded in the form of a supplement:
https://www.ocean-sci-discuss.net/os-2017-89/os-2017-89-AC5-supplement.pdf

---

## Author Comment (AC7) · 15 Mar 2018

The comment was uploaded in the form of a supplement:
https://www.ocean-sci-discuss.net/os-2017-89/os-2017-89-AC7-supplement.pdf

---

## Author Response (AR3)

**Author's Response**

The manuscript describes the diel vertical migration of macrozooplankton in Patagonian fjords based on a comprehensive data set. Although the scientific approach is not novel, the quality of the presented results is good and the results are well presented. I found several typos (see below) and encourage the authors to check the language of the manuscript carefully. In my opinion, Figure 10 does not say much and should be deleted. Otherwise I suggest to publish the manuscript in Ocean Science after minor revision.

We eliminated figure 10 from the text.

**Minor comments:**
L38 northern Patagonian fjord

We changed "north Patagonian Fjord" by northern Patagonian Fjord
L57 interchange with what?

We have edited the paragraph "Turbulence induced by tidal flow over the sill apparently enhances the interchange of nutrients and oxygen concentrations with the surface layer, creating a productive environment for many marine species, where prey-predator relationship might be favored."

L58 please rephrase "fruitful environment!

We replaced "fruitful" by "productive"
L100 … and these turbulent eddies …

We changed the sentence "and turbulent eddies" by "and these turbulent eddies"
L104 will be addressed

We replaced "will be address" by "will be addressed"
L105 (e.g., temperature, …)

We replaced "e.g., Temperature, …" by "e.g., temperature, …"
L112 "Dead Zones"

We replaced "Dead zones" by "Dead Zones"
L128 Patagonia

We replaced "Patagonian" by "Patagonia"

L160 no comma after et al.

We eliminated comma.
L161/162 (0-10 m, 10-25 m, 25-50 m and 50-100 m)

We replaced "(200-50 m, 50-25 m, 25-10 m and 10-0 m depth)" by "(0-10 m, 10-25 m, 25-50 m and 50-100 m)"

L204 favors

We replaced "favor" by "favors"

L205 deepens

We replaced "deepen" by "deepens"
L221 fjord systems

We replaced "Fjord System" by "fjord system"
L257 reference not in the reference list, please check

We added the reference to the reference list.
"IOC, SCOR and IAPSO: The international thermodynamic equation of seawater - 2010: Calculation and use of thermodynamic properties. Intergovernmental Oceanographic Commission, Manuals and Guides No. 56, UNESCO, 196 pp, 2010".

L305 third ADCP

We replaced "thrid" by "third"

L310 are shown in Figures 5, 7 and 8

We replaced "were showed in figures 5, 7 and 8" by "are shown in Figures 5, 7 and 8"

L314 with 614.4 kHz frequency (…) was moored …

We replaced "was 614.4 kHz frequency (…) was moored …" by "with 614.4 kHz frequency (…) was moored …"

L331 no comma after et al.

We eliminated comma.
L347 scatterers. (no comma)

We eliminated comma.
L393: zooplankter's (?)

We replaced "zooplankter's" by "zooplankton species"
L402 no comma after et al.

We eliminated comma.
L462 due to the presence

We replaced "due to presence" by "due to the presence"
L487 survey data

We replaced "surveys data" by "survey data"
L480 was moderate. (I am not impressed by the correlation)

We replaced "was high" by "was moderate"

L506 macrozooplankton

We replaced "macrozooplankton groups" by "macrozooplankton"
L532 turbulent kinetic energy (TKE)

We clarified the sentence
L542 regime

We replaced "regimen" by "regime"

Delete Figure 10

We eliminated figure 10 from the text.